# Trust Region Inverse Reinforcement Learning: Explicit Dual Ascent using Local Policy Updates

**Anish Diwan** [1 2]  **Davide Tateo** [1 3]  **Christopher E. Mower** [4]  **Haitham Bou Ammar** [4 5]  **Jan Peters** [1 6 7 2]
**Oleg Arenz** [1 2]

## Abstract

Inverse reinforcement learning (IRL) is typically formulated as maximizing entropy subject to matching the distribution of expert trajectories. Classical (dual-ascent) IRL guarantees monotonic performance improvement but requires fully solving an RL problem each iteration to compute dual gradients. More recent adversarial methods avoid this cost at the expense of stability and monotonic dual improvement, by directly optimizing the primal problem and using a discriminator to provide rewards. In this work, we bridge the gap between these approaches by enabling monotonic improvement of the reward function and policy without having to fully solve an RL problem at every iteration. Our key theoretical insight is that a trust-region-optimal policy for a reward function update can be globally optimal for a smaller update in the same direction. This smaller update allows us to explicitly optimize the dual objective while only relying on a local search around the current policy. In doing so, our approach avoids the training instabilities of adversarial methods, offers monotonic performance improvement, and learns a reward function in the traditional sense of IRL—one that can be globally optimized to match expert demonstrations. Our proposed algorithm, *Trust Region Inverse Reinforcement Learning (TRIRL)*, outperforms state-of-the-art imitation learning methods across multiple challenging tasks by a factor of 2.4x in terms of aggregate inter-quartile mean, while recovering reward functions that generalize to system dynamics shifts. [‡]

[‡] Code and supplementary material are available at: https://anishhdiwan.github.io/trust-region-irl/. [1]Technical University of Darmstadt [2]Robotics Institute Germany (RIG) [3]Lund University [4]Huawei, Noah's Ark Lab [5]University College London [6]hessian.AI [7]German Research Center for AI (DFKI). Correspondence to: Anish Diwan <anish.diwan@robot-learning.de>.

*Proceedings of the 43rd International Conference on Machine Learning*, Seoul, South Korea. PMLR 306, 2026. Copyright 2026 by the author(s).

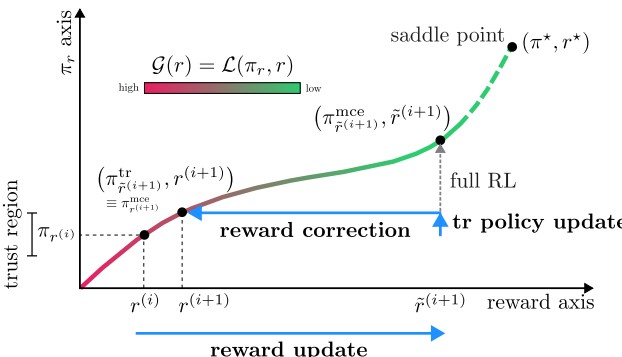

*Figure 1.* A comparison of TRIRL (ours) and a MaxCausalEnt-IRL style update. The Lagrangian dual to be optimized is indicated by the curve $\mathcal{L}(\pi_r, r)$. MCE-IRL performs a full RL optimization after updating the reward function. In contrast, TRIRL only optimizes the policy within a trust region of the previous MCE policy and accounts for this by correcting the updated reward function. Trust region policy updates are much cheaper to compute and reward correction ensures that the new policy-reward pair gets closer to the saddle point of $\mathcal{L}(\pi, r)$. Our algorithm hence has the same monotonic improvement guarantees as MCE-IRL, while being able to converge in complex, high-dimensional settings.

## 1. Introduction

As Autonomous agents become prevalent in everyday settings, empowering these systems with human-like behaviour becomes an important challenge. This challenge can be addressed with Inverse Reinforcement Learning (IRL) (Ng & Russell, 2000; Russell, 1998), a machine learning framework where agents infer the underlying intentions of human demonstrations in the form of a reward function. By optimizing this learnt reward with reinforcement learning, IRL methods can recover robust imitation policies with the additional benefit of transferring the behaviour to new environments. However, inferring an informative reward function from demonstrations is challenging, and therefore, many methods focus on Imitation Learning (Osa et al., 2018), the problem of directly learning a policy that matches the demonstrations in some given environment.

IL only addresses the policy-side problem in IRL and does not extract the underlying reward function. It is generally solved using adversarial optimization, with methods such

as GAIL (Ho & Ermon, 2016) and its several variants (Peng et al., 2018; Kostrikov et al., 2019; Ghasemipour et al., 2020; Peng et al., 2021) being the standard approach. These methods formulate IL as a two-player min-max game, where one player (called the discriminator) assigns local rewards based on how well the RL policy imitates the expert, while the policy updates itself to maximise this local signal given by the discriminator. However, adversarial IL is inherently unstable and noisy because the discriminator reward provides only a local correction signal. Practically speaking, these methods are challenging to tune and their performance is highly task-dependent. This raises a key question: *how can we learn informative reward functions and effective policies in a scalable, principled way, while avoiding the instability of adversarial IL?* We approach this question by retracing modern adversarial IL back to its theoretical roots in the Maximum Causal Entropy IRL (MCE-IRL) framework (Ziebart et al., 2008; 2010). Our insights motivate a scalable approach for imitation learning and inverse reinforcement learning based on explicit dual ascent.

MCE-IRL interprets imitation as having similar occupancy[1] over the underlying Markov Decision Process. An imitation policy is considered MCE optimal if it has the maximum causal entropy among all candidate policies that induce the same occupancy as the expert. This problem is typically solved by optimizing its Lagrangian $\mathcal{L}(\pi, r)$, where $\mathcal{G}(r) = \mathcal{L}(\pi_r, r)$ is its dual, and $\pi_r$ is the optimal policy for $r$. The original MCE-IRL method (Ziebart et al., 2010) is a dual-ascent algorithm that *alternates between policy optimization and reward updates*. At every iteration, the reward function's (dual variable) gradients are given by the difference between expert and agent feature counts and the policy (primal variable) is learnt by solving the maximum entropy RL problem till convergence. Modern adversarial IL algorithms reformulate this dual-ascent procedure as a two-player min-max game. However, in doing so, the global intermediate reward function from MCE-IRL is replaced by a local reward based on the previous policy's rollouts (provided by a discriminator). Consequently, the policy update is not MCE optimal; instead, the policy only takes a few gradient steps toward maximizing the entropy-augmented reward. While this adversarial procedure has the same saddle point as MCE-IRL (Ho & Ermon, 2016, Proposition 3.2), it relies on per-step local optimization to obtain a solution and does not optimize the reward function corresponding to the dual. Ultimately, the local discriminator rewards and suboptimal policies render it practically unstable and difficult to train reliably and consistently.

This paper proposes a new, non-adversarial IRL algorithm that solves the problem of local rewards and suboptimal

intermediate policies. We present *Trust Region Inverse Reinforcement Learning (TRIRL)*, an algorithm that performs dual ascent on the original MCE-IRL problem, leading to monotonic performance improvement and stable learning. Crucially, our method avoids both the need to run expensive, full RL solutions at every time step (like MCE-IRL) or the reliance on approximate local optimization (like GAIL).

Our work builds on prior work by Arenz et al. (2016) that showed that instead of descending $\mathcal{G}(r)$ along its parametric gradient, it is significantly more efficient to perform a reward update in function space. Given an initial max-ent pair $(\pi_r, r)$ and their function-space reward update on $r$, our main result is to show that policy optimization within a reverse KL trust-region around $\pi_r$ is sufficient to find a policy $\pi^{\mathrm{mce}}$ that is max-ent optimal for a new, corrected reward function computed by taking a *smaller update step* along the same function-space update direction. Hence, we can leverage a novel mechanic for IRL: instead of finding a max-ent optimal policy for the updated reward, we find a trust region optimal policy for this reward, and correct the reward function to account for the fact that our policy was only optimized locally. This results in a valid maximum entropy pair $(\pi^{\mathrm{mce}}_{r_{\mathrm{corrected}}}, r_{\mathrm{corrected}})$. This means that the policy we learn at each iteration (i) is a global optimizer of the corrected reward function and (ii) gets closer to the entropy-regularized expert occupancy in terms of the reverse KL divergence. By repeating this procedure we recover an IRL algorithm with the same monotonic performance improvement as MCE-IRL. We illustrate our method in Figure 1. Our algorithm outperforms prior works like GAIL, AIRL, AMP, LSIQ, NEAR, and SFM (Ho & Ermon, 2016; Fu et al., 2018; Peng et al., 2021; Al-Hafez et al., 2023a; Diwan et al., 2025; Jain et al., 2025) on a variety of challenging continuous control tasks.

**Related Work.** Behavioural Cloning (BC) is arguably the most straightforward approach to Imitation Learning. It formulates IL as a supervised learning problem to find a policy that closely matches the demonstrated actions. Although BC methods like Pomerleau (1991); Reddy et al. (2019) have previously shown successful imitation capabilities, the supervised fitting has theoretical limitations—namely covariate shift, poor generalization, and the need for large datasets—that degrade their performance in real-world environments. On the other hand, Inverse RL methods like MCE-IRL (Ziebart et al., 2008; 2010) learn imitation policies using reinforcement learning and are hence more robust than BC. This formulation can also be used for deriving direct IL methods, such as GAIL (Ho & Ermon, 2016), which reformulate the dual-ascent formulation of Ziebart et al. (2010) into an adversarial min-max optimization procedure that minimizes the Jensen-Shannon divergence between the agent and expert occupancies. Several other works

---

[1]Originally, Ziebart et al. (2010) formalize MCE-IRL as a state-visitation matching problem but this is equivalent to occupancy matching (Ho & Ermon, 2016; Arenz et al., 2016).

build on this adversarial formulation. For instance, Fu et al. (2018) modify the GAIL procedure into a state-based algorithm focusing on reward recovery. Ghasemipour et al. (2020) reformulate it for general $f$-divergences and Peng et al. (2018) leverage the empirical benefits of $\chi^2$-divergence GANs (Mao et al., 2017) for adversarial IL. Kostrikov et al. (2020) present ValueDice, a method that leverages the inverse Bellman operator to reformulate GAIL into a value-function based off-policy, distribution matching approach. Several other works (Kostrikov et al., 2019; Orsini et al., 2021; Diwan et al., 2025) shed light on important factors like survival/termination bias induced by learnt reward functions, empirical training dynamics of adversarial IL, and theoretical reasoning for their instability. While the adversarial IL prior works listed here are diverse in terms of their contributions, all of them are prone to instabilities rooted in the local rewards and suboptimal policies arising from adversarial learning. The work of Ziebart et al. (2010) can also be formulated into non-adversarial techniques. Arenz & Neumann (2020) extend ValueDICE by formulating a lower-bound reward function for reverse KL distribution matching and using soft actor critic (SAC) (Haarnoja et al., 2018) to learn a $Q$-function and policy. Garg et al. (2021) propose IQ-Learn, a similar method that generalizes to a variety of divergences between the agent and expert occupancies. Instead of optimizing the dual and primal variables separately, IQ-learn learns a $Q$-function via distribution matching and approximates the policy using a closed-form relationship to the $Q$-function. However, IQ-learn requires a dynamics model to recover the reward function from this learnt $Q$-function. Recently, Al-Hafez et al. (2023a) introduced LSIQ, an extension of IQ-Learn that leverages the benefits of $\chi^2$-divergence minimization and uses mixture distributions for improved performance. Several of these non-adversarial IRL methods still use off-policy RL (SAC) for policy learning. This makes it challenging to apply them to larger, highly parallelized environments, where SAC faces scaling challenges. Finally, Boularias et al. (2011) previously explored the idea of constraining policy updates in IRL to a relative-entropy-based trust region around a baseline policy. However, their method does not address reward updates under local policy optimization, and inherits the scaling challenges of Ziebart et al. (2010), requiring trajectory-level sampling, and hand-specified features.

## 2. Background

**Preliminaries.** Similarly to previous work, we model the environment as a Markov Decision Process (MDP) defined by a tuple $(\mathcal{S}, \mathcal{A}, \mu_0, \mathcal{P}, r, \gamma)$, where $\mathcal{S}$ is the state space, $\mathcal{A}$ is the action space, $\mu_0$ is the initial state distribution, $\mathcal{P}(\mathbf{s}'|\mathbf{s}, \mathbf{a})$ represents the transition dynamics, $r(\mathbf{s}, \mathbf{a}) \in \mathbb{R}$ is the (unknown) reward function, and $\gamma$ is the discount factor. $\Pi$ is the set of all stationary stochastic policies

mapping states in $\mathcal{S}$ to actions in $\mathcal{A}$. We define the occupancy measure $\rho_\pi(\mathbf{s}, \mathbf{a}) = \pi(\mathbf{a}|\mathbf{s}) \sum_{t=0}^{\infty} \gamma^t \mu_t^\pi(\mathbf{s})$, where $\mu_t^\pi(\mathbf{s}') = \sum_{\mathbf{s}} \mu_t^\pi(\mathbf{s}) \sum_{\mathbf{a}} \pi(\mathbf{a}|\mathbf{s}) P(\mathbf{s}'|\mathbf{s}, \mathbf{a})$ is the state distribution for $t > 0$, with $\mu_0^\pi(\mathbf{s}) = \mu_0(\mathbf{s})$. We work in the $\gamma$-discounted infinite horizon setting and use an expectation with respect to a policy $\pi \in \Pi$ to denote an expectation with respect to the trajectory it generates. We refer to the (unknown) expert policy as $\pi_E$, its occupancy measure as $\rho_E(\mathbf{s}, \mathbf{a})$, and the dataset of expert demonstrations as $\mathcal{D}$. Following prior work (Ziebart et al., 2010; Haarnoja et al., 2018), we define entropy regularized value functions $V_\pi(\mathbf{s}) = \mathbb{E}_\pi [Q_\pi(\mathbf{s}, \mathbf{a}) - \log \pi(\mathbf{a}|\mathbf{s})]$ and $Q_\pi(\mathbf{s}, \mathbf{a}) = R(\mathbf{s}, \mathbf{a}) + \gamma \mathbb{E}_{\mathbf{s}' \sim \mathcal{P}} [V_\pi(\mathbf{s}')]$ as "soft" value functions.

**Max Causal Entropy IRL/RL.** The maximum entropy principle (Jaynes, 1957) states that given several probability distributions consistent with the observed data, the best approach is to choose the least committal one (i.e., one that has the maximum entropy). When applied to the problem of (inverse) reinforcement learning (Ziebart et al., 2010; Eysenbach & Levine, 2022), the maximum entropy principle offers improved robustness and an elegant solution to the problem of encouraging exploration. Given a set of demonstration trajectories $\mathcal{D} = \{(\mathbf{s}_i, \mathbf{a}_i)\}_{i=1}^N$ sampled from an expert, MCE-IRL aims to find a reward function $r \in \mathbb{R}^{\mathcal{S} \times \mathcal{A}}$ and a policy $\pi \in \Pi$ that solve the optimization problem: $\max_\pi \min_r (\mathbb{E}_{\rho_\pi}[r(\mathbf{s}, \mathbf{a})] + H(\pi)) - \mathbb{E}_{\rho_E}[r(\mathbf{s}, \mathbf{a})]$. Whereas, given a reward function $r$, Maximum Entropy RL aims to find a policy $\pi$ that maximises the expected reward plus entropy: $\max_\pi \mathbb{E}_{\rho_\pi}[r(\mathbf{s}, \mathbf{a})] + H(\pi)$. Here $H(\pi) \triangleq \mathbb{E}_\pi[-\log \pi(\mathbf{a}|\mathbf{s})]$ is the discounted causal entropy of the policy in state $\mathbf{s}$. On solving the Lagrangian, the optimal policy satisfies the Boltzmann distribution $\pi^\star(\mathbf{a}|\mathbf{s}) = \frac{1}{Z_\mathbf{s}} \exp (Q^\star(\mathbf{s}, \mathbf{a}))$ where

$$Q^\star(\mathbf{s}, \mathbf{a}) = r(\mathbf{s}, \mathbf{a}) + \gamma \mathbb{E}_{\mathbf{s}' \sim \mathcal{P}} \left[ \log \sum_{\mathbf{a}'} \exp(Q^\star(\mathbf{s}', \mathbf{a}')) \right],$$

$$V^\star(\mathbf{s}) = \log \sum_{\mathbf{a}'} \exp(Q^\star(\mathbf{s}', \mathbf{a}'))$$

are the optimal soft value functions and $Z_\mathbf{s}$ is a normalization term. Notice that maximum entropy RL is a subroutine of the MCE-IRL procedure.

**IRL by Distribution Matching.** Our work is based on IRL by reverse KL-divergence based distribution matching. This problem is deeply rooted in prior work (Arenz et al., 2016; Fu et al., 2018; Kostrikov et al., 2020; Arenz & Neumann, 2020; Ghasemipour et al., 2020; Garg et al., 2021), and is expressed by the optimization problem,

$$\max_{\pi(\mathbf{a}|\mathbf{s})} \mathbb{E}_{\rho_\pi(\mathbf{s})} [H(\pi(\mathbf{a}|\mathbf{s}))] - \beta \mathbb{E}_{\rho_\pi(\mathbf{s}, \mathbf{a})} \left[ \log \frac{\rho_\pi(\mathbf{s}, \mathbf{a})}{\rho_E(\mathbf{s}, \mathbf{a})} \right] \quad (1)$$

where $\beta$ trades off between entropy regularization and the objective of matching the expert's occupancy. Arenz et al. (2016) show that under this problem setting, in the limit $\beta \to \infty$, the optimal policy and value functions are the same as for MCE-IRL, and the reward function for matching the expert's distribution is, $r^\star = \beta \log \left( \rho_E(\mathbf{s},\mathbf{a}) / \rho_{\pi^\star}(\mathbf{s},\mathbf{a}) \right)$, where $(\pi^\star, r^\star)$ is the same saddle point as in MCE-IRL. The optimal reward function depends on the state-action distribution induced by the optimal policy of Equation (1), resulting in a cyclic dependency between the optimal policy and the reward function. However, Arenz et al. (2016) show that the reward function can also be learnt by iteratively applying the function-space update operator $\mathcal{U}_\epsilon^{\rho^{(i)}}$

$$r^{(i+1)} = \left( \mathcal{U}_\epsilon^{\rho^{(i)}} \right) r^{(i)} = r^{(i)} - \epsilon \underbrace{\left( r^{(i)} - \beta \log \frac{\rho_E(\mathbf{s},\mathbf{a})}{\rho_{\pi^{(i)}}(\mathbf{s},\mathbf{a})} \right)}_{\delta^{(i)}}$$
(2)

to the current estimate of the reward function, $r^{(i)}$, where $\pi^{(i)}$ is the optimal maximum entropy policy for reward function $r^{(i)}$ (we use $\rho^{(i)}$ as shorthand for $\rho_{\pi^{(i)}}$). It can be shown that this update aligns with the gradient of the dual of Equation (1) (proof in Appendix A) and is empirically several orders of magnitude more efficient than using the actual gradient. Hence, successive applications of Equation (2) lead to monotonic improvement in the objective. Such prior work was restricted to linear approximations of the system dynamics and assumed the expert distribution to be Gaussian. Under these relaxations, the optimal policy can be computed using dynamic programming and the log density ratio, $\log \left( \rho_E(\mathbf{s},\mathbf{a}) / \rho_{\pi^{(i)}}(\mathbf{s},\mathbf{a}) \right)$, computed in closed form. However, such relaxations do not work in most real-world applications.

## 3. Trust Region Inverse Reinforcement Learning

In real-world, continuous settings, the densities $\rho_\pi(\mathbf{s}, \mathbf{a})$ and $\rho_E(\mathbf{s}, \mathbf{a})$ cannot be computed in closed form. Instead, the log density ratio, $\log \left( \rho_E(\mathbf{s},\mathbf{a}) / \rho_{\pi^{(i)}}(\mathbf{s},\mathbf{a}) \right)$ needs to be approximated from the dataset of expert demonstrations and policy rollouts. In such a setting, the distribution matching approach yields a reward formulation that is very similar to modern adversarial techniques like GAIL (Ho & Ermon, 2016). However, as highlighted in Section 1, the local reward approximations and suboptimal policy updates often lead to unstable convergence for such adversarial methods.

Instead, we propose non-adversarial reward function updates similar to Equation (2) that ensure monotonic improvement on the objective function (Equation (1)). The main challenge in applying Equation (2) in real-world settings is that this update—just like the gradient-based update—depends on the optimal policy $\pi^{(i)}(\mathbf{a}|\mathbf{s})$ for the current re-

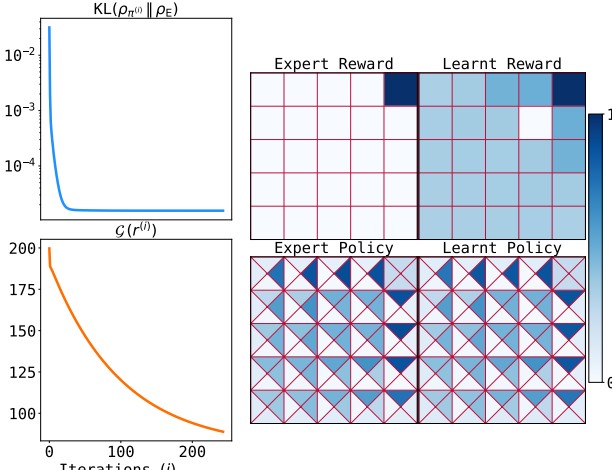

*Figure 2.* We demonstrate TRIRL in a grid-world experiment and compare policies and normalized rewards. We also show the monotonically improving reverse KL divergence and dual objective. TRIRL exactly recovers the expert's policy and recovers a reward function that matches the expert's reward (except for ambiguity due to the temporal credit assignment problem).

ward function estimate $r^{(i)}(\mathbf{s}, \mathbf{a})$. Vanilla MCE-IRL finds this optimal policy by running maximum entropy reinforcement learning until convergence. However, solving the full deep reinforcement learning problem at every iteration of inverse reinforcement learning is prohibitively expensive for most real-world applications. Ideally, we would like to come up with a procedure that obtains the current iteration's optimal policy only after a few updates to the previous policy, such that we could iteratively keep obtaining the MCE optimal policy and the corresponding updated reward function. In this paper, we derive reward function updates similar to Equation (2) and show that the MCE-optimal policy can be computed based on trust-region optimal policies

$$\pi_{\text{tr}}^{(i+1)} = \arg\max_{\pi(\mathbf{a}|\mathbf{s})}$$
$$\mathbb{E}_{\rho_\pi(\mathbf{s})} \left[ H(\pi(\mathbf{a}|\mathbf{s})) \right] + \mathbb{E}_{\rho_\pi(\mathbf{s})} \left[ \mathbb{E}_\pi \left[ r^{(i)}(\mathbf{s},\mathbf{a}) \right] \right],$$
$$\text{s.t.} \quad \mathbb{E}_{\rho_\pi(\mathbf{s})} \left[ \text{KL} \left( \pi || \pi_{\text{tr}}^{(i)} \right) \right] \leq \zeta.$$
(3)

Our main contribution is to show that a trust region policy $\pi_{\text{tr}}^{(i+1)}(\mathbf{a}|\mathbf{s})$ for a reward function $\tilde{r}^{(i+1)} = \left( \mathcal{U}_\epsilon^{\rho^{(i)}} \right) r^{(i)}$ is the maximum-entropy optimal policy with respect to a different reward function $r^{(i+1)} = \left( \mathcal{U}_{\epsilon_{\text{tr}}}^{\rho^{(i)}} \right) r^{(i)}$, where reward updates are computed as per Equation (2) and $\epsilon_{\text{tr}} \leq \epsilon$. This concretely solves the problem of finding an optimal policy in the inner loop of IRL since finding a trust-region optimal policy is easier and only requires a local search. Further, the rewards learnt using Equation (2) are guaranteed to improve the distribution matching objective's dual. Maximizing this

reward function guarantees that the updated policy's occupancy is closer to the entropy-regularized expert occupancy than the occupancy of the policy used to learn this reward (in terms of reverse KL divergence).

**Lemma 3.1.** *The trust-region optimal maximum causal entropy policy for reward function $r(\mathbf{s}, \mathbf{a})$, corresponds to the optimal maximum causal entropy policy for a reward function that takes the form $r_\eta(\mathbf{s}, \mathbf{a}) = \frac{1}{(1+\eta)} r(\mathbf{s}, \mathbf{a}) + \frac{\eta}{(1+\eta)} \log \pi^{(i)}(\mathbf{a}|\mathbf{s})$, where $\eta$ is the Lagrangian multiplier corresponding to the trust-region constraint.*

**Theorem 3.2.** *Let $\pi_{tr}^{(i+1)}(\mathbf{a}|\mathbf{s})$ denote a trust-region optimal policy for a reward function $\tilde{r}^{(i+1)} = (\mathcal{U}_\epsilon^{\rho^{(i)}}) r^{(i)}$ with stepsize $\epsilon$. There exists a positive stepsize $\epsilon_{tr} \leq \epsilon$, such that $\pi_{tr}^{(i+1)}(\mathbf{a}|\mathbf{s})$ is an optimal maximum causal entropy policy with respect to the reward function $r^{(i+1)} = (\mathcal{U}_{\epsilon_{tr}}^{\rho^{(i)}}) r^{(i)}$.*

Proofs for Lemma 3.1 and Theorem 3.2 are in Appendix A. Assuming we know an optimal policy $\pi^{(i)}(\mathbf{a}|\mathbf{s})$ for the reward function $r^{(i)}(\mathbf{s}, \mathbf{a})$, Theorem 3.2 shows that $\pi_{\text{tr}}$ on $\left(\mathcal{U}_\epsilon^{\rho^{(i)}}\right) r^{(i)} \equiv \pi_{\text{MCE}}$ on $\left(\mathcal{U}_{\epsilon_{\text{tr}}}^{\rho^{(i)}}\right) r^{(i)}$. Here $\epsilon_{\text{tr}} = \epsilon/(1+\eta)$ is a step size smaller than the initial step size on which the trust-region optimal policy was computed, and $\eta \geq 0$ is the Lagrangian multiplier associated with the trust region constraint in Equation (3). Theorem 3.2 can be used to construct an IRL procedure where we first use a large step size to get the reward function $\tilde{r}^{(i+1)}$, then compute a trust region optimal policy using this, and subsequently correct the updated reward function using a new step size $\epsilon_{\text{tr}}$. The corrected reward function is such that the previously computed trust region optimal policy is MCE-optimal on it. Figure 1 illustrates this procedure and compares it with maximum causal entropy IRL. Starting from constant rewards and a uniform policy ($r^{(0)}$ ; $\pi^{(0)}$), this subroutine can be iterated to monotonically improve on the dual objective solely based on trust-region updates on the policy. Algorithm 1 shows an overview of *Trust Region Inverse Reinforcement Learning (TRIRL)* and Figure 2 demonstrates it in a discrete setting.

# 4. Practical Considerations

In this section, we discuss practical considerations needed for applying TRIRL to real-world problems. These involve estimating the log density ratio used in the reward update (Equation (2)), realizing function-space updates on parametric reward functions, and performing the trust-region policy updates. Furthermore, we discuss important properties, namely the ability to learn from observations and to re-optimize the reward function after changes to the system dynamics.

---

**Algorithm 1** Trust Region Inverse RL

1: **Initialize:** $\epsilon$ ; $r^{(0)} = 0.0$ ; $\pi^{(0)} = \text{unif.}$
2: **Output:** $r^\star$ and $\pi^\star$
3: **repeat**
4:     rollout $\pi^{(i)}$ ; learn $D^{(i)} \approx \log\left(\rho_E / \rho_{\pi^{(i)}}\right)$
5:     $\tilde{r}^{(i+1)} = (1-\epsilon) r^{(i)} + \epsilon\beta D^{(i)}$  Equation (2)
6:     $\pi_{\text{tr}}^{(i+1)}$ & $\eta^{(i+1)} \leftarrow$ trust region policy update
7:     $\epsilon_{\text{tr}} = \epsilon/(1+\eta^{(i+1)})$
8:     $r^{(i+1)} = (1-\epsilon_{\text{tr}}) r^{(i)} + \epsilon_{\text{tr}}\beta D^{(i)}$
9:     $\pi_{\text{tr}}$ on $\tilde{r}^{(i+1)} \equiv \pi_{\text{MCE}}$ on $r^{(i+1)}$
10:    $r^{(i)} \leftarrow r^{(i+1)}$ ; $\pi^{(i)} \leftarrow \pi_{\text{tr}}^{(i+1)}$
11: **until** converged

---

## 4.1. Density Ratio Estimation

In continuous state spaces, the log density ratio $\log\left(\rho_E(\mathbf{s}, \mathbf{a}) / \rho_{\pi^{(i)}}(\mathbf{s}, \mathbf{a})\right)$ cannot be computed analytically and must be approximated using a neural network. Following previous works, we train a binary classifier $D$ to distinguish between expert and agent samples. Given infinite data and an optimally trained $D$, the log density ratio is equivalent to the logits of $D$ (Menon & Ong, 2016). While adversarial methods like GAIL directly use $\log \sigma(D(.))$ as a reward function, we use the logits to perform a function-space reward update. Consequently, unlike adversarial methods, we do not require a fine-tuned balance between the classifier and policy updates. Where adversarial IRL methods typically fail with a perfect discriminator (Diwan et al., 2025), our method always benefits from having a highly accurate classifier. An accurate classifier further improves the practical accuracy of the monotonic improvement guarantees of Equation (2), wherein the reward is a global solution of the MCE-IRL Lagrangian and the policy is a global optimizer of this reward.

## 4.2. Circular Buffer of Discriminators

An additional challenge arises from the fact that we perform updates in function space rather than parameter space. While the functional update has been shown to be more efficient than parametric gradient descent (Arenz et al., 2016), the recursive application of the update in Equation (2) results in a reward function that explicitly depends on the entire history of trained discriminators. Maintaining this history is computationally prohibitive, particularly regarding the memory footprint. Furthermore, imposing inductive biases (e.g., action invariance) on the aggregate reward function would require enforcing them on every individual discriminator.

To circumvent these limitations, one could project the reward function onto a parametric model at every iteration via a regression problem. However, such projections introduce approximation errors that may affect training stability. In-

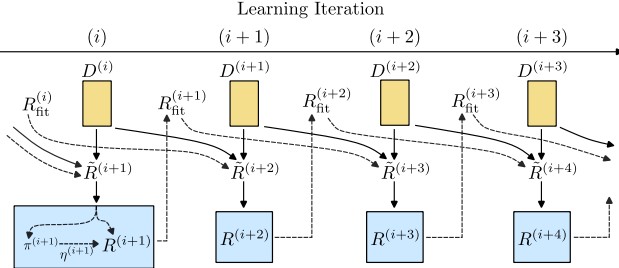

**Figure 3.** An illustration of a discriminator buffer of size $k = 2$. Given a fitted reward $R_{\text{fit}}^{(i-k)}$ and discriminators $\{D^{(i)}\}_{i-k}^{i}$, intermediate uncorrected rewards $\tilde{R}^{(i+1)}$ are computed by repeated application of line 8 in Algorithm 1. Then, a trust region policy $\pi^{(i+1)}$ is learnt, and the final corrected reward $R^{(i+1)}$ is computed using line 5 in Algorithm 1.

stead, we propose a middle ground: we maintain a fixed buffer of the $k$ most recent discriminators along with a parametric reward function $R_{\text{fit}}^{(i-k)}$ that was fitted at iteration $i - k$. Due to the exponential decay of past coefficients caused by the recursive updates, the approximation errors of the fitted reward function have limited effect on the overall rewards, allowing us to impose structural priors on the parametric model without sacrificing the precision of the most recent functional updates.

We illustrate the discriminator buffer in Figure 3. While this does entail added computational effort, these operations are highly parallelizable and are completed relatively quickly by leveraging jit-compilation. An evaluation of the effects on runtime and memory consumption can be found in Appendix B.1.

### 4.3. Trust-Region Optimal Policy Updates

Notice that Theorem 3.2 only necessitates reward correction based on multipliers $\eta$ and a policy that is optimal w.r.t the trust region corresponding to $\eta$. However, there is no additional constraint on the actual trust region bound to which $\eta$ corresponds. Hence, Theorem 3.2 can be applied naively by treating $\eta$ as a hyperparameter and enforcing the trust region constraint under expectation through an auxiliary loss $\mathcal{L}_{\text{tr}} \triangleq \eta \, \text{KL}\left(\pi || \pi_{\text{tr}}^{(i)}\right)$. We note that it is theoretically sound to optimize such an expected KL penalty (instead of a hard constraint).

However, specifying $\eta$ arbitrarily is difficult since its effect depends on the scale of the reward function. Instead, it is more convenient to explicitly specify a trust region and strictly enforce it for better training stability. To ensure such explicit trust region satisfaction, and an $\eta$ corresponding to the current optimization landscape, we propose to leverage differentiable trust-region projection layers (Otto et al., 2021, TRPL). TRPL parametrizes the policy as a state-dependent Gaussian distribution $\pi(\cdot|\mathbf{s}) = \mathcal{N}(\mu_\theta(\mathbf{s}), \Sigma_\phi)$

and exactly enforces the analytical reverse KL trust region by the application of a projection layer during policy optimization. The projection layer maps every violating policy prediction back into the trust region such that the projected policy parameters $\left(\tilde{\mu}_{\mathbf{s}}, \tilde{\Sigma}\right)$ satisfy a trust-region constraint around the previous policy while simultaneously minimizing the distance to policy predictions $(\mu_\theta(\mathbf{s}), \Sigma_\phi)$. TRPL uses separate bounds for the mean and covariance and obtains Lagrangian multipliers $\eta_\mu(\mathbf{s}) \geq 0$ and $\eta_\Sigma \geq 0$. However, as we require a single, scalar multiplier $\eta$, we compute the maximum over the state-dependent multipliers for mean and covariance, $\eta = \max_{\text{batch}}\{\eta_\mu(\mathbf{s}), \eta_\Sigma\}$, resulting in a more conservative step size. While these design choices enable us to strictly enforce trust-region satisfaction, we note that this variant slightly deviates from our theory, in that the trust-region constraint is satisfied for every state individually, instead of enforcing it in expectation as in Equation (3). For further details on TRPL, we refer to Appendix C.3 and the original work (Otto et al., 2021). Both variants, (i) treating $\eta$ as penalty coefficient (*TR loss* version) and (ii) computing it using TRPL ($\max \eta$ version) use PPO for policy optimization.

We further evaluate a variant based on TRPL with an interesting update mechanic: instead of using the maximum eta over all states, we use state-dependent step sizes—if a larger $\eta$ is required to satisfy a trust-region in a given state, this variant will result in smaller changes to the rewards of that state. In combination with the circular buffer of discriminators, we can even use this mechanism to adapt step sizes in hindsight. Here, we need to additionally keep track of past policies to recompute past step sizes based on the given state. We will refer to this variant as *retrospective-$\eta$*.

### 4.4. Learning from Observations & Transfer Learning

Finally, we highlight that TRIRL is a general framework and can accommodate a variety of discriminator architectures. TRIRL can be used in observation-based imitation settings, where we do not have access to the states and actions of the expert, but only to observed features. As a special case, it can be used for state-only observations, potentially using a discriminator based on state transitions, $D(\mathbf{s}, \mathbf{s}')$ (Torabi et al., 2018). Further, TRIRL directly learns reward functions that can be globally optimized, enabling us to re-optimize the policy to adapt to changes in the system dynamics. This distinguishes it from AIRL (Fu et al., 2018) and related methods, where the reward function is "entangled" with the log-density of the learnt policy (the advantage function). We demonstrate learning from observations through motion-capture imitation experiments on humanoid robots, and global, transferable reward learning in Section 5.1.

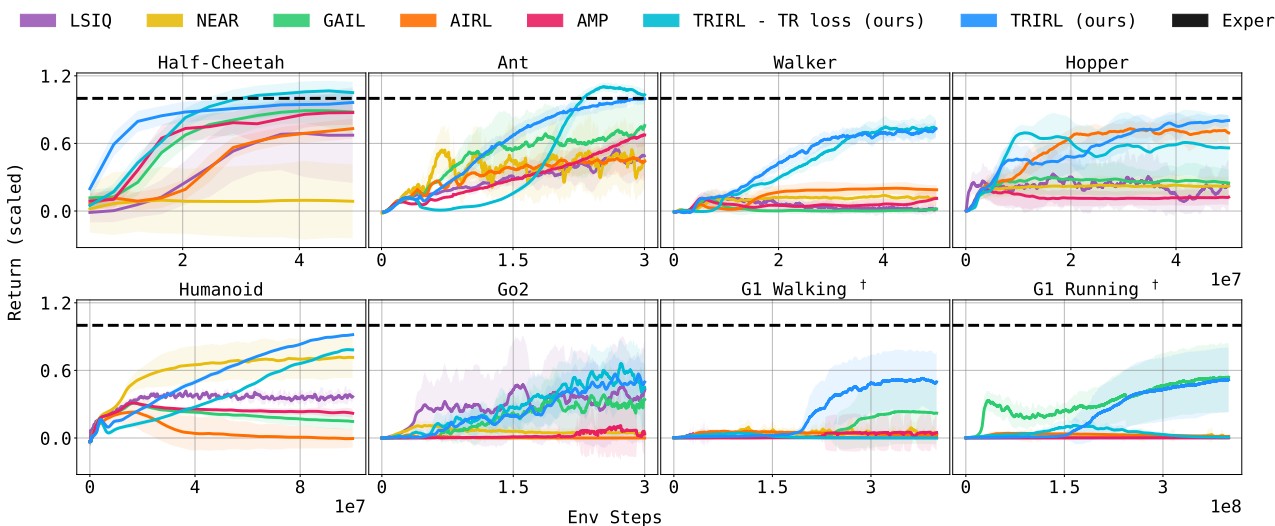

*Figure 4.* Imitation learning results on Mujoco benchmarks and robotics tasks. [†] The G1 tasks use mocap demonstrations where only the expert's observations are available.

## 5. Experiments

Through our experiments and ablation studies, we aim to answer the following questions:

1. How does TRIRL compare to prominent prior works in complex imitation learning settings?

2. Is there any advantage to computing Lagrangian multipliers retrospectively? What is the impact of reward fitting, the TR loss, TRPL, and the discriminator buffer on performance?

3. Can TRIRL learn a global reward function? Is this reward also transferable?

We conduct experiments on continuous control tasks on the following Mujoco benchmarks: Half-Cheetah, Ant, Walker, Hopper, Humanoid; as well as more complex robotics settings: Unitree G1 Walking/Running, Unitree Go2 Locomotion. A PPO policy trained till convergence is used as the expert and 30 demonstration trajectories are collected per task. In the Unitree G1 environments, we use the LocoMujoco (Al-Hafez et al., 2023b) motion capture dataset and train using a state-based discriminator $D(\mathbf{s}, \mathbf{s}')$. The motion capture datasets for walking and running contain approximately 35 and 9 trajectories, respectively (1000 step horizon). We compare TRIRL with the following prior works: GAIL (Ho & Ermon, 2016), AIRL (Fu et al., 2018), AMP (Peng et al., 2021), LSIQ (Al-Hafez et al., 2023a), NEAR (Diwan et al., 2025), and SFM (Jain et al., 2025). Together, these baselines enable a systematic comparison of our method against adversarial and non-adversarial approaches, state-based and state-only variants, as well as methods focused on reward recovery. All methods are tuned separately for each task and we report the mean performance across 20 independent seeds. Due to space constraints, we defer secondary results (ablations, runtime/memory comparisons, data scaling) to Appendices B and B.1, and provide experimental details and baseline descriptions in Appendix C.

Figure 4 shows that TRIRL matches and, in most cases, outperforms all baselines in all environments. Further, the TR loss version of TRIRL—which is theoretically sound— also often outperforms most baselines, albeit performing poorly on harder robotics tasks like the G1 due to its weaker trust region enforcement. Next, we evaluate the relative significance of the components of TRIRL by modifying its policy optimization and reward correction steps. We carry out ablation experiments on the same Mujoco benchmarks and compare the following ablated configurations:

| | |
|---|---|
| (i) $\max \eta$ | (ii) TR loss |
| (iii) retrospective $\eta$ | (iv) retrospective $\eta$ w/o reward fitting |
| (v) w/o disc. buffer | (vi) w/o TRPL & disc. buffer |
| (vii) GAIL w/ TRPL | |

Figure 5 leads to several interesting observations. First, we find that in most cases, the $\max \eta$ variant of TRIRL outperforms all other variants. While this variant slightly deviates from our theory, the strict trust region enforcement of TRPL and the associated non-arbitrary lagrangian multipliers, indeed offer increased stability and performance. Moreover, we see that retrospectively computing state-dependent Lagrangian multipliers usually does not yield significant performance improvements. It turns out that taking a $\max$ over previously computed Lagrangian multipliers is typically

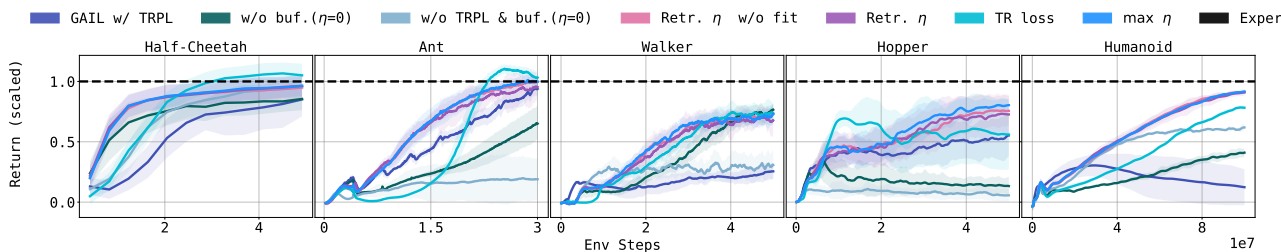

*Figure 5.* An ablation study comparing the relative performance of TRIRL's variants on Mujoco benchmarks.

*Table 1.* A comparison of normalized performance during training, retraining from scratch with the learnt reward, and retraining from scratch with the learnt reward on an environment with changed dynamics. For both baselines, we find that only a fraction of the seeds converged with any meaningful performance (notice the high variance). This is likely because of different training - retraining initialization, and their reward being only local. The global reward learnt by our method, however, ensures convergence much more reliably. [†] For NEAR, training and re-training performance are equivalent.

| Task | Training | | | Retraining | | | Transfer | | |
|------|---------|------|------|-----------|------|------|---------|------|------|
| | TRIRL | AIRL | NEAR | TRIRL | AIRL | NEAR | TRIRL | AIRL | NEAR |
| Point Maze | $\mathbf{1.03 \pm 0.01}$ | $0.45 \pm 0.12$ | $0.28 \pm 0.09$ | $\mathbf{0.98 \pm 0.01}$ | $0.35 \pm 0.07$ | $0.28 \pm 0.09$ | $\mathbf{0.96 \pm 0.001}$ | $0.06 \pm 0.64$ | $0.29 \pm 0.13$ |
| Ant | $\mathbf{0.91 \pm 0.17}$ | $0.59 \pm 0.25$ | $0.46 \pm 0.29$ | $\mathbf{0.63 \pm 0.09}$ | $0.10 \pm 0.13$ | $0.46 \pm 0.29$ | $\mathbf{0.89 \pm 0.12}$ | $0.42 \pm 0.25$ | $0.33 \pm 0.18$ |
| Half Cheetah | $\mathbf{0.83 \pm 0.19}$ | $0.39 \pm 0.14$ | $0.09 \pm 0.28$ | $\mathbf{0.70 \pm 0.24}$ | $0.08 \pm 0.28$ | $0.09 \pm 0.28$ | (W) $\mathbf{0.63 \pm 0.29}$ (MG) $\mathbf{0.30 \pm 0.13}$ | (W) $0.16 \pm 0.25$ (MG) $-0.10 \pm 0.06$ | (W) $0.10 \pm 0.18$ (MG) $-0.06 \pm 0.12$ |
| Hopper | $0.49 \pm 0.16$ | $\mathbf{0.68 \pm 0.11}$ | $0.22 \pm 0.09$ | $\mathbf{0.36 \pm 0.13}$ | $0.12 \pm 0.11$ | $0.22 \pm 0.09$ | — | — | — |

conservative enough to ensure trust region satisfaction on a new rollout batch. In this ablation, we also examine how fitting the reward function onto a parametric model affects performance. As expected, such fitting has a small effect on imitation learning performance. Further, we observe that the TR loss variant of our method also performs well in Mujoco benchmarks. This version, being less computationally demanding, is a reasonable alternative in such simpler settings. Finally, ablated configurations (v), (vi), and (vii) show the contribution of the discriminator buffer and the trust region projection layer. Configuration (v) removes reward correction by replacing TRIRL's reward function with an *uncorrected* interpolation of a buffer of discriminators (with $\eta = 0$). Configuration (vi) just boils down to GAIL with the same *uncorrected* rewards. Configuration (vii) is GAIL with TRPL for policy optimization. The poor performance of these ablated configurations shows that our method's improvements aren't simply rooted in TRPL (Otto et al., 2021) or the buffer of discriminators. Instead, we attribute TRIRL's improvements to the explicit optimization of the dual objective.

### 5.1. Global & Transferable Rewards

Here, we show that TRIRL learns a global reward function, i.e., one that can be re-optimized from scratch, starting from a new random agent initialization. Figure 2 demonstrates this in a discrete setting. In continuous settings, we rely on function approximation to learn rewards. In these environments, we demonstrate global reward learning by freezing a trained reward network and re-optimizing it from scratch with PPO. Crucially, this retraining is done on a new set of

seeds to ensure that the agent is initialized differently than during training. The same setup is also used to evaluate the transferability of the learnt reward. Ideally, a global reward function captures the expert's intrinsic motivations (e.g. moving forward), rather than rewarding the agent just for duplicating the specific state transitions executed by the expert. Such a reward should also transfer to a different environment with similar goals. We test this by re-optimizing the learnt reward on an environment with changed dynamics. Specifically, we use the Point Maze Flipped and Ant Disabled environments from (Fu et al., 2018), where the maze is flipped and the dynamics of the Ant task are changed by shortening and disabling the agent's front legs. We further add two new transfer tasks, Half Cheetah Windy (W) and Half Cheetah Mars Gravity (MG), where a constant 5m/s wind blows against the agent, and the gravitational constant is changed to $3.73m/s^2$. We evaluate a feature-based variant of our method with a feature encoder, and a reward function that is linear in these features. Table 1 shows comparisons against AIRL (Fu et al., 2018) and NEAR (Diwan et al., 2025), both of which claim to learn a global reward function. TRIRL outperforms baselines in both retraining and transfer settings, while ensuring the best training performance.

## 6. Discussion and Outlook

We show that trust region policy updates (primal optimization), followed by reward (dual) correction, result in monotonic improvement of the distribution matching IRL objective. Given this, we present *Trust Region Inverse Reinforce-*

*ment Learning (TRIRL)*, a non-adversarial IRL method that achieves monotonic improvement of the reward function and policy, without having to solve a full RL problem at each iteration of IRL (like classical IRL methods). Practically, our method offers stable training characteristics and is capable of recovering global reward functions that are robust to changes in system dynamics.

While our method outperforms strong baselines, there are a few limitations and failure modes to consider. First, we note that the theoretical guarantees discussed in our work do not perfectly translate to real-world settings with function approximation. Discriminator-based density ratio estimation can introduce approximation errors due to data and sampling limitations, and using a discriminator buffer further increases VRAM requirements—though its impact is minimal as shown in Appendix B. We also clarify that, although our experiments use TRPL to learn trust-region policies, our method is not limited to Gaussian policies or to TRPL's computationally expensive trust-region update. In practice, solving the MaxEnt RL problem in Algorithm 1 only requires a likelihood-based policy so that the entropy can be evaluated. Our theory (Theorem 3.2) provides a general framework for learning MCE-optimal IRL policies, and we also present a trust-region loss variant of our method that often outperforms the baselines. Finally, the IRL problem is ill-posed due to temporal credit assignment: different reward functions may induce the same optimal policy in a given environment, but lead to different behavior under different dynamics (Ng et al., 1999). Like prior work, our method is also subject to this limitation. However, in Appendix B, we briefly highlight that this issue can be addressed by instilling inductive biases into the problem, which prior work such as (Ho & Ermon, 2016; Fu et al., 2018) cannot accommodate because it requires access to specific expert states/actions. In terms of failure modes, our method could fail to converge with very lax trust region bounds and take prohibitively long with very conservative ones. In the TR loss version, where $\eta$ is a hyperparameter, this is also not intuitively tunable. Future work in such trust region based IRL methods could focus on extending this procedure to general $f$-divergences, and studying the properties of function-space reward updates in detail.

## Software and Data

To aid reproducibility, the full algorithm and ablation studies are given in Appendix B, hyperparameters are listed in Appendix C.5, and specific implementation details are provided in Appendices C.2 to C.4. Code & supplementary material are available at https://anishhdiwan.github.io/trust-region-irl/.

## Acknowledgements

This research has been partially supported by the German Research Foundation DFG within RTG 2761 LokoAssist (Grant no. 450821862) and INTENTION (Grant no. 506123304), and the German Federal Ministry of Research, Technology and Space (BMFTR) under the Robotics Institute Germany (RIG). This project has been supported by a hardware donation by NVIDIA through the Academic Grant Program. Calculations for this research were conducted on the Lichtenberg high-performance computer of the TU Darmstadt.

## Impact Statement

This paper presents work whose goal is to advance the field of Machine Learning. There are many potential societal consequences of our work, none of which we feel must be specifically highlighted here.

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

# A. Proofs

We first list some fundamental properties of our optimization problem that are necessary for our analysis.

*Fact* A.1 (Policy-Occupancy Relationship). There is a unique relationship between a policy $\pi(\mathbf{a}|\mathbf{s})$ and it's state-action occupancy $\rho_\pi(\mathbf{s}, \mathbf{a})$, wherein $\rho_\pi(\mathbf{s}, \mathbf{a}) = \pi(\mathbf{a}|\mathbf{s})\rho_\pi(\mathbf{s})$. Proof in (Ho & Ermon, 2016, Lemmas 3.2, 3.3) and (Syed et al., 2008, Theorem 2).

*Fact* A.2 (Entropy Concavity). The expected entropy $\mathbb{E}_{\rho_\pi(\mathbf{s})}\left[H(\pi(\mathbf{a}|\mathbf{s}))\right]$ is concave in $\rho_\pi(\mathbf{s}, \mathbf{a})$. Proof in (Ho & Ermon, 2016; Neu et al., 2017, Lemma 3.1, Appendix A.1).

*Fact* A.3 (Strong Duality). The optimization problem in Equation (3) has strong duality. The proof of this naturally follows from Fact A.2 and the strong duality of entropy regularized RL (Peters et al., 2010; Achiam et al., 2017; Neu et al., 2017; Geist et al., 2019). The expected entropy term is concave, the reward term is linear, the KL constraint is convex, and the Bellman flow constraints are linear (all in $\rho_\pi(\mathbf{s}, \mathbf{a})$). Hence Slater's conditions hold and strong duality applies. Finally from Fact A.1, the policy is equivalently obtained by optimizing over $\rho_\pi(\mathbf{s}, \mathbf{a})$ followed by marginalization.

***Proof of reward update soundness***. Here, we prove that the reward update $r^{(i+1)} = \left(\mathcal{U}_\epsilon^{\rho^{(i)}}\right) r^{(i)} = r^{(i)} - \epsilon\delta^{(i)}$ (Equation (2)) corresponds to an ascent direction with respect to the dual of Equation (1). This proof is a reworking of the results from Arenz et al. (2016). To show this, we first redefine the reward update in terms of it's parameters and arrive at an equivalent form of the original update direction. Then, we show that this aligns with the gradient of the dual of Equation (1).

From (Ziebart et al., 2010; Arenz et al., 2016), we note that the learnt reward is linear in it's features. Let $\rho_E(\mathbf{s}, \mathbf{a}) = Z_E^{-1} \exp\left(\phi_E \, \psi(\mathbf{s}, \mathbf{a})\right)$ and $\rho_{\pi^{(i)}}(\mathbf{s}, \mathbf{a}) = Z_{\rho^{(i)}}^{-1} \exp\left(\phi_{\rho^{(i)}} \, \psi(\mathbf{s}, \mathbf{a})\right)$ be Boltzmann distributions with energy functions that are—without loss of generality—linear in features $\psi(\mathbf{s}, \mathbf{a})$ that are not assumed to be known. In our analysis, we denote reward parameters $\theta$, occupancy parameters $\phi$, and a general feature function $\psi(\mathbf{s}, \mathbf{a})$. Estimates of the optimal reward and optimal policy occupancy are termed $\hat{r}$ and $\rho_{\hat{\pi}}$. Owing to the reward's linearity, we replace the above update with an equivalent version in parameter space:

$$\theta^{(i+1)} = \theta^{(i)} - \epsilon\delta^{(i)} \quad \text{where} \quad \delta^{(i)} = \theta^{(i)} - \beta \log \frac{\rho_E(\mathbf{s}, \mathbf{a})}{\rho_{\pi^{(i)}}(\mathbf{s}, \mathbf{a})}$$

and $\hat{r} = \beta \log \frac{\rho_E(\mathbf{s}, \mathbf{a})}{\rho_{\pi^{(i)}}(\mathbf{s}, \mathbf{a})}$ is an estimate of the optimal reward function $r^\star$. Arenz et al. (2016) also show that this can be used to obtain an estimate of the optimal policy's occupancy:

$$\rho_{\hat{\pi}}(\mathbf{s}, \mathbf{a}) \propto \exp\left(\log \rho_E(\mathbf{s}, \mathbf{a}) - \frac{1}{\beta}\hat{r}\right)$$
$$\propto \exp\left(\left(\phi_E - \frac{\theta^{(i)}}{\beta}\right)^\mathsf{T} \psi(\mathbf{s}, \mathbf{a})\right)$$
$$\propto \exp\left(\hat{\phi}^\mathsf{T} \psi(\mathbf{s}, \mathbf{a})\right) \tag{4}$$

Then, the estimate of the optimal reward is given by

$$\beta \log \frac{\rho_E(\mathbf{s}, \mathbf{a})}{\rho_{\pi^{(i)}}(\mathbf{s}, \mathbf{a})} = \beta \left(\log Z_E^{-1} \exp\left(\phi_E \, \psi(\mathbf{s}, \mathbf{a})\right) - \log Z_{\rho^i}^{-1} \exp\left(\phi_{\rho^i} \, \psi(\mathbf{s}, \mathbf{a})\right)\right)$$
$$= \beta\left(\phi_E - \phi_{\rho^i}\right) + \text{const.}$$

The update direction then translates to:

$$\delta^{(i)} = \theta^{(i)} - \beta\left(\phi_E - \phi_{\rho^i}\right)$$
$$= \beta\left(\frac{\theta^{(i)}}{\beta} - \phi_E + \phi_{\rho^i}\right)$$
$$= \beta\left(\phi_{\rho^i} - \hat{\phi}\right) \quad \text{(from Equation (4))}. \tag{5}$$

Given Equation (5), we now show that $\delta' = 1/\beta \, \delta^{(i)} = \phi_{\rho^i} - \hat{\phi}$ aligns with the gradient of the Lagrangian dual $\mathcal{G}$. Where,

$$\mathcal{G} = \mathbb{E}_{\mu_0}\left[V_\pi^{\text{soft}}(\mathbf{s}_0)\right] + \beta \log \sum_{(\mathbf{s},\mathbf{a})} \exp\left(\log \rho_E(\mathbf{s},\mathbf{a}) - \frac{1}{\beta} r^\star(\mathbf{s},\mathbf{a})\right)$$

$$\frac{\partial G}{\partial \theta} = \mathbb{E}_{\rho_\pi(\mathbf{s})}\left[\psi(\mathbf{s},\mathbf{a})\right] - \mathbb{E}_{\rho_{\hat{\pi}}}\left[\psi(\mathbf{s},\mathbf{a})\right]$$

We refer the reader to Arenz et al. (2016) for the derivations of the Lagrangian dual and it's gradient. Let $\langle . , . \rangle$ define the inner product of two vectors.

$$\begin{aligned}
\left\langle \delta' , \frac{\partial G}{\partial \theta} \right\rangle &= \left\langle \phi_{\rho^i} - \hat{\phi} , \mathbb{E}_{\rho_\pi(\mathbf{s})}\left[\psi(\mathbf{s},\mathbf{a})\right] - \mathbb{E}_{\rho_{\hat{\pi}}}\left[\psi(\mathbf{s},\mathbf{a})\right] \right\rangle \\
&= \mathbb{E}_{\rho_\pi(\mathbf{s})}\left[\left(\phi_{\rho^i} - \hat{\phi}\right)^\mathsf{T} \psi(\mathbf{s},\mathbf{a})\right] - \mathbb{E}_{\rho_{\hat{\pi}}}\left[\left(\phi_{\rho^i} - \hat{\phi}\right)^\mathsf{T} \psi(\mathbf{s},\mathbf{a})\right] \\
&= \mathbb{E}_{\rho_\pi(\mathbf{s})}\left[\log \frac{\rho_{\pi^{(i)}}(\mathbf{s},\mathbf{a})}{\rho_{\hat{\pi}}(\mathbf{s},\mathbf{a})} + \log \frac{Z_{\rho^i}}{Z_{\hat{\rho}}}\right] + \mathbb{E}_{\rho_{\hat{\pi}}}\left[\log \frac{\rho_{\hat{\pi}}(\mathbf{s},\mathbf{a})}{\rho_{\pi^{(i)}}(\mathbf{s},\mathbf{a})} + \log \frac{Z_{\hat{\rho}}}{Z_{\rho^i}}\right] \\
&= D_{\text{KL}}\left[\rho_{\pi^{(i)}} \,||\, \hat{\rho}\right] + D_{\text{KL}}\left[\hat{\rho} \,||\, \rho_{\pi^{(i)}}\right] \\
&\geq 0 \quad \text{(sum of KLs)}
\end{aligned}$$

The update direction $\delta^{(i)}$ must align with the gradient since their inner product is positive. In other words, repeated applications of Equation (2) lead to monotonic improvement in the objective. $\qquad\square$

*Proof of Lemma 3.1.* From Fact A.3, we note the existence of an optimal Lagranginan multiplier. We further clarify that we do not require exact, per-state satisfaction of the constraint in Equation (3). Instead, it is sufficient to optimize an expected KL constraint using an $\eta$-weighted penalty term (as shown below). We start with the Lagrangian expression of optimization problem 3, given by

$$\mathcal{L}(\pi,\eta) = \mathbb{E}_{\rho_\pi(\mathbf{s})}\left[H(\pi(\mathbf{a}|\mathbf{s}))\right] + \mathbb{E}_{\rho_\pi(\mathbf{s})}\left[\mathbb{E}_\pi\left[r(\mathbf{s},\mathbf{a})\right]\right] + \eta\zeta - \eta\mathbb{E}_{\rho_\pi(\mathbf{s})}\left[\text{KL}\Big(\pi(\mathbf{a}|\mathbf{s})||\pi^{(i)}(\mathbf{a}|\mathbf{s})\Big)\right]$$

For a given Lagrangian multiplier $\eta$, the optimal policy can be computed as

$$\begin{aligned}
\pi^\eta(\mathbf{a}|\mathbf{s}) =& \arg\max_{\pi(\mathbf{a}|\mathbf{s})} \mathcal{L}(\pi,\eta) \\
=& \arg\max_{\pi(\mathbf{a}|\mathbf{s})} \mathbb{E}_{\rho_\pi(\mathbf{s})}\left[H(\pi(\mathbf{a}|\mathbf{s}))\right] + \mathbb{E}_{\rho_\pi(\mathbf{s})}\left[\mathbb{E}_\pi\left[r(\mathbf{s},\mathbf{a})\right]\right] + \eta\zeta - \eta\mathbb{E}_{\rho_\pi(\mathbf{s})}\left[\text{KL}\Big(\pi(\mathbf{a}|\mathbf{s})||\pi^{(i)}(\mathbf{a}|\mathbf{s})\Big)\right] \\
=& \arg\max_{\pi(\mathbf{a}|\mathbf{s})} \mathbb{E}_{\rho_\pi(\mathbf{s})}\left[H(\pi(\mathbf{a}|\mathbf{s})) + \mathbb{E}_\pi\left[r(\mathbf{s},\mathbf{a})\right] - \eta\text{KL}\Big(\pi(\mathbf{a}|\mathbf{s})||\pi^{(i)}(\mathbf{a}|\mathbf{s})\Big)\right] + \eta\zeta \\
=& \arg\max_{\pi(\mathbf{a}|\mathbf{s})} \mathbb{E}_{\rho_\pi(\mathbf{s})}\left[H(\pi(\mathbf{a}|\mathbf{s})) + \mathbb{E}_\pi\left[r(\mathbf{s},\mathbf{a})\right] - \eta\mathbb{E}_\pi\left[\log \frac{\pi(\mathbf{a}|\mathbf{s})}{\pi^{(i)}(\mathbf{a}|\mathbf{s})}\right]\right] + \eta\zeta \\
=& \arg\max_{\pi(\mathbf{a}|\mathbf{s})} \mathbb{E}_{\rho_\pi(\mathbf{s})}\left[H(\pi(\mathbf{a}|\mathbf{s})) + \mathbb{E}_\pi\left[r(\mathbf{s},\mathbf{a})\right] - \eta\mathbb{E}_\pi\left[\log \pi(\mathbf{a}|\mathbf{s}) - \log \pi^{(i)}(\mathbf{a}|\mathbf{s})\right]\right] + \eta\zeta \\
=& \arg\max_{\pi(\mathbf{a}|\mathbf{s})} \mathbb{E}_{\rho_\pi(\mathbf{s})}\left[H(\pi(\mathbf{a}|\mathbf{s})) + \mathbb{E}_\pi\left[r(\mathbf{s},\mathbf{a})\right] - \eta\mathbb{E}_\pi\left[\log \pi(\mathbf{a}|\mathbf{s})\right] + \eta\mathbb{E}_\pi\left[\log \pi^{(i)}(\mathbf{a}|\mathbf{s})\right]\right] + \eta\zeta \\
=& \arg\max_{\pi(\mathbf{a}|\mathbf{s})} \mathbb{E}_{\rho_\pi(\mathbf{s})}\left[(1+\eta)H(\pi(\mathbf{a}|\mathbf{s})) + \mathbb{E}_\pi\left[r(\mathbf{s},\mathbf{a}) + \eta\log \pi^{(i)}(\mathbf{a}|\mathbf{s})\right]\right] + \eta\zeta \\
=& \arg\max_{\pi(\mathbf{a}|\mathbf{s})} (1+\eta)\mathbb{E}_{\rho_\pi(\mathbf{s})}\left[H(\pi(\mathbf{a}|\mathbf{s})) + \mathbb{E}_\pi\left[\frac{r(\mathbf{s},\mathbf{a})}{(1+\eta)} + \frac{\eta}{(1+\eta)}\log \pi^{(i)}(\mathbf{a}|\mathbf{s})\right]\right] + \eta\zeta \\
=& \pi_{r_\eta}^{\text{MCE}}
\end{aligned}$$

$\qquad\square$

***Proof of Theorem 3.2.***

$$\pi_{\text{tr}}^{(i+1)}(\mathbf{a}|\mathbf{s})$$

$$= \underset{\pi(\mathbf{a}|\mathbf{s})}{\arg\max} \, \mathbb{E}_{\rho_\pi(\mathbf{s})} \left[ H(\pi(\mathbf{a}|\mathbf{s})) \right] + \mathbb{E}_{\rho_\pi(\mathbf{s})} \left[ \mathbb{E}_\pi \left[ \tilde{r}^{(i+1)}(\mathbf{s}, \mathbf{a}) \right] \right]$$

$$- \eta \mathbb{E}_{\rho_\pi(\mathbf{s})} \left[ \text{KL}\left( \pi || \pi_{\text{tr}}^{(i)} \right) \right] + \eta \zeta$$

*Apply Lemma 3.1*

$$= \underset{\pi(\mathbf{a}|\mathbf{s})}{\arg\max} \, \mathbb{E}_{\rho_\pi(\mathbf{s})} \left[ H(\pi(\mathbf{a}|\mathbf{s})) + \mathbb{E}_\pi \left[ \frac{\tilde{r}^{(i+1)}(\mathbf{s}, \mathbf{a})}{(1+\eta)} + \frac{\eta}{(1+\eta)} \log \pi^{(i)}(\mathbf{a}|\mathbf{s}) \right] \right]$$

*Substitute $\tilde{r}^{(i+1)} = \left( \mathcal{U}_\epsilon^{\rho^{(i)}} \right) r^{(i)}$*

$$= \underset{\pi(\mathbf{a}|\mathbf{s})}{\arg\max} \, \mathbb{E}_{\rho_\pi(\mathbf{s})} \left[ H(\pi(\mathbf{a}|\mathbf{s})) + \mathbb{E}_\pi \left[ \frac{1}{(1+\eta)} \left( (1-\epsilon) r^{(i)}(\mathbf{s}, \mathbf{a}) + \epsilon \beta \log \frac{\rho_E(\mathbf{s}, \mathbf{a})}{\rho_{\pi^{(i)}}(\mathbf{s}, \mathbf{a})} \right) \right. \right.$$

$$\left. \left. + \frac{\eta}{(1+\eta)} \log \pi^{(i)}(\mathbf{a}|\mathbf{s}) \right] \right]$$

*$\pi^{(i)}(\mathbf{a}|\mathbf{s})$ is Boltzmann*

$$= \underset{\pi(\mathbf{a}|\mathbf{s})}{\arg\max} \, \mathbb{E}_{\rho_\pi(\mathbf{s})} \left[ H(\pi(\mathbf{a}|\mathbf{s})) + \mathbb{E}_\pi \left[ \frac{1}{(1+\eta)} \left( (1-\epsilon) r^{(i)}(\mathbf{s}, \mathbf{a}) + \epsilon \beta \log \frac{\rho_E(\mathbf{s}, \mathbf{a})}{\rho_{\pi^{(i)}}(\mathbf{s}, \mathbf{a})} \right) \right. \right.$$

$$\left. \left. + \frac{\eta}{(1+\eta)} \left( Q^{(i)}(\mathbf{s}, \mathbf{a}) - V^{(i)}(\mathbf{s}) \right) \right] \right]$$

$$= \underset{\pi(\mathbf{a}|\mathbf{s})}{\arg\max} \, \mathbb{E}_{\rho_\pi(\mathbf{s})} \left[ H(\pi(\mathbf{a}|\mathbf{s})) + \mathbb{E}_\pi \left[ \frac{1}{(1+\eta)} \left( (1-\epsilon) r^{(i)}(\mathbf{s}, \mathbf{a}) + \epsilon \beta \log \frac{\rho_E(\mathbf{s}, \mathbf{a})}{\rho_{\pi^{(i)}}(\mathbf{s}, \mathbf{a})} \right) \right. \right.$$

$$\left. \left. + \frac{\eta}{(1+\eta)} \left( r^{(i)}(\mathbf{s}, \mathbf{a}) + \gamma \mathbb{E}_{\mathbf{s}'} \left[ V^{(i)}(\mathbf{s}') \right] - V^{(i)}(\mathbf{s}) \right) \right] \right]$$

$$= \underset{\pi(\mathbf{a}|\mathbf{s})}{\arg\max} \, \mathbb{E}_{\rho_\pi(\mathbf{s})} \left[ H(\pi(\mathbf{a}|\mathbf{s})) + \mathbb{E}_\pi \left[ \frac{(1-\epsilon)}{(1+\eta)} r^{(i)}(\mathbf{s}, \mathbf{a}) + \frac{\epsilon}{(1+\eta)} \beta \log \frac{\rho_E(\mathbf{s}, \mathbf{a})}{\rho_{\pi^{(i)}}(\mathbf{s}, \mathbf{a})} \right. \right.$$

$$\left. \left. + \frac{\eta}{(1+\eta)} r^{(i)}(\mathbf{s}, \mathbf{a}) + \frac{\eta}{(1+\eta)} \left( \gamma \mathbb{E}_{\mathbf{s}'} \left[ V^{(i)}(\mathbf{s}') \right] - V^{(i)}(\mathbf{s}) \right) \right] \right]$$

*Policy invariance under potential shaping (Ng et al., 1999)*

$$= \underset{\pi(\mathbf{a}|\mathbf{s})}{\arg\max} \, \mathbb{E}_{\rho_\pi(\mathbf{s})} \left[ H(\pi(\mathbf{a}|\mathbf{s})) + \mathbb{E}_\pi \left[ \frac{(1-\epsilon)}{(1+\eta)} r^{(i)}(\mathbf{s}, \mathbf{a}) + \frac{\epsilon}{(1+\eta)} \beta \log \frac{\rho_E(\mathbf{s}, \mathbf{a})}{\rho_{\pi^{(i)}}(\mathbf{s}, \mathbf{a})} + \frac{\eta}{(1+\eta)} r^{(i)}(\mathbf{s}, \mathbf{a}) \right] \right]$$

$$= \underset{\pi(\mathbf{a}|\mathbf{s})}{\arg\max} \, \mathbb{E}_{\rho_\pi(\mathbf{s})} \left[ H(\pi(\mathbf{a}|\mathbf{s})) + \mathbb{E}_\pi \left[ \left( 1 - \frac{\epsilon}{(1+\eta)} \right) r^{(i)}(\mathbf{s}, \mathbf{a}) + \frac{\epsilon}{(1+\eta)} \beta \log \frac{\rho_E(\mathbf{s}, \mathbf{a})}{\rho_{\pi^{(i)}}(\mathbf{s}, \mathbf{a})} \right] \right]$$

$$= \underset{\pi(\mathbf{a}|\mathbf{s})}{\arg\max} \, \mathbb{E}_{\rho_\pi(\mathbf{s})} \left[ H(\pi(\mathbf{a}|\mathbf{s})) + \mathbb{E}_\pi \left[ \left( \mathcal{U}_{\left( \frac{\epsilon}{(1+\eta)} \right)}^{\rho^{(i)}} \right) r^{(i)} \right] \right]$$

$$= \pi_{r^{(i+1)}}^{\text{MCE}}$$

□

## B. Additional Results

In this section, we present auxiliary experimental results that support our claims. Table 2 compares our method with Successor Feature Matching (SFM) (Jain et al., 2025). Table 3 compares aggregate normalized inter-quartile mean and VRAM usage. Figure 6 compares wall-clock training times between TRIRL and baselines in Mujoco benchmarks and robotics imitation learning settings. As described further in Appendix C.2, we reimplement all methods (including baselines) to leverage jit-compilation in Jax and use a parallelized RL simulator to further reduce training times. We find that adversarial IL baselines like GAIL (Ho & Ermon, 2016), AIRL (Fu et al., 2018), and AMP (Peng et al., 2021) train faster in comparison to TRIRL. The primary computational costs of our method arise from two components: (i) the reward correction step that uses a buffer of discriminators, and (ii) TRPL for policy optimization (which uses a numerical solver). The runtime impact of the reward correction step is illustrated by comparing adversarial methods with the TR loss variant of TRIRL (Section 4.3). Although this variant incurs a modest increase in runtime, it already outperforms most baselines in terms of performance. Among the TRIRL variants, the TR loss version is also the fastest because of its simpler policy update. This is followed by the $\max \eta$ and retrospective $\eta$ versions. NEAR (Diwan et al., 2025) is roughly comparable to TRIRL because of the high computational burden of learning an energy-based reward via score matching. In our experiments, LSIQ (Al-Hafez et al., 2023a) takes much longer to train because we intentionally reduce RL parallelization in favour of improved SAC performance. The limited ability to scale with parallelized environments is a key limitation of SAC-based imitation learning methods and ultimately constrains their applicability to complex environments such as the Unitree G1. Figures 7 and 8 highlight the improved training stability offered by our method. We find that our method is also much more robust to overtrained discriminators. Finally, Algorithm 2 shows an extend, practical pseudocode of our method and Figure 9 shows the learnt reward function in the point maze task.

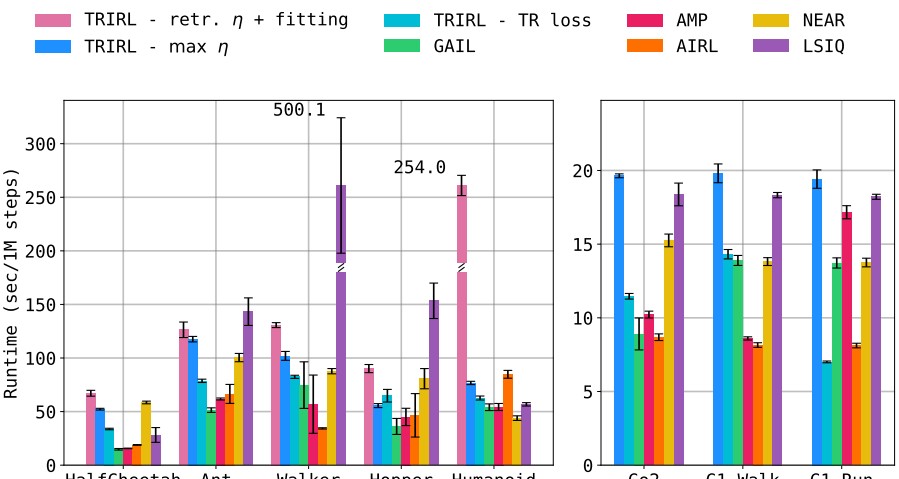

*Figure 6.* A comparison of the runtimes of all methods. Except for LSIQ on the G1 tasks, all other methods were trained on an RTX 3090 GPU. An RTX PRO 6000 Blackwell was used to accommodate the large replay buffer and batch sizes needed for LSIQ on the complex G1 environment (Appendix C.5).

**TRIRL in Discrete Settings:** Here, we briefly explain the procedure used to generate Figure 2. In the discrete case, our implementation closely follows Algorithm 1. The log density ratio is computed by evaluating the expert's and the agent's occupancies analytically from the policy (and transition dynamics). The reward update and reward correction are also computed analytically for the whole grid at once. We use soft value iteration for policy optimization and augment the reward with a reverse KL divergence trust region penalty weighted by a scheduled $\eta$. Soft value iteration returns Boltzmann policies and guarantees convergence to the trust region optimal policy. In the discrete setting, we naturally also don't need discriminators or a discriminator buffer.

*Table 2.* We compare our method against Successor Feature Matching (SFM) ([Jain et al., 2025](#)) with a forward dynamics model (FDM) for base features and a TD7 policy (best reported configuration in the paper). We used the official SFM codebase, tuned SFM following the guidelines in the paper, and trained it till convergence. Learning successor features in our complex robotics environments (Go2/G1) requires substantial engineering effort, which we leave for upcoming future work.

| Task | **TRIRL** $\max \eta$ | **TRIRL TR loss** | **SFM** |
|------|------|------|------|
| Ant | $\mathbf{1.00 \pm 0.05}$ | $1.03 \pm 0.06$ | $0.36 \pm 0.07$ |
| Half Cheetah | $\mathbf{1.05 \pm 0.04}$ | $0.96 \pm 0.04$ | $\mathbf{1.07 \pm 0.07}$ |
| Walker | $\mathbf{0.73 \pm 0.08}$ | $0.72 \pm 0.08$ | $0.51 \pm 0.21$ |
| Hopper | $\mathbf{0.81 \pm 0.06}$ | $0.56 \pm 0.29$ | $0.56 \pm 0.22$ |
| Humanoid | $\mathbf{0.92 \pm 0.03}$ | $0.78 \pm 0.05$ | $0.70 \pm 0.08$ |

*Table 3.* Aggregate normalized inter-quartile mean (IQM) with 95% CI (higher is better) and aggregate VRAM usage (lower is better) across all imitation learning tasks. [†]: LSIQ has higher memory usage in our experiments because of the larger replay buffers. SFM is omitted due to only being evaluated in Mujoco environments (where it underperforms our method).

| Algorithm | IQM | 95 % CI low | 95 % CI high | Memory (Gb) |
|------|------|------|------|------|
| TRIRL $\max \eta$ (ours) | **0.7881** | 0.7786 | 0.7972 | 7.5774 |
| TRIRL - TR loss (ours) | **0.6293** | 0.5794 | 0.6796 | 7.5774 |
| GAIL | 0.3297 | 0.2725 | 0.3868 | 1.1192 |
| LSIQ | 0.1795 | 0.1494 | 0.2134 | 3.6924 [†] |
| AIRL | 0.1694 | 0.1467 | 0.1928 | 1.2744 |
| AMP | 0.1167 | 0.0985 | 0.1418 | *1.0105* |
| NEAR | 0.0986 | 0.0818 | 0.1184 | 2.5355 |

**Feature-Based Variants:**    As highlighted in Section [4.4](#), our method is a general framework and is not limited to a specific discriminator architecture or data modality. We can hence instil additional inductive biases in our method by doing IRL in the space of features. For example, it is reasonable to assume that the reward function for humanoid locomotion is a function of the floating-base velocity, torso height, and action cost. We can compute such features from the provided expert demonstrations, and learn non-linear extensions into a useful feature-space. TRIRL can directly learn reward functions in such feature spaces. We find that such inductive biases can greatly improve the the quality of learnt rewards (Figure [9](#)) and highlight that this is generally a limitation of prior works like ([Fu et al., 2018](#); [Garg et al., 2021](#)). We use this strategy for our experiments in Section [5.1](#), where we learn a linear reward function in the space of non-linear features of the environment's state-only base features. We learn these non-linear features by modelling a feature encoder, predicting the energy function of encoded features, and minimizing the denoising score matching loss ([Song & Ermon, 2019](#), Equation 5) to perturbed expert samples.

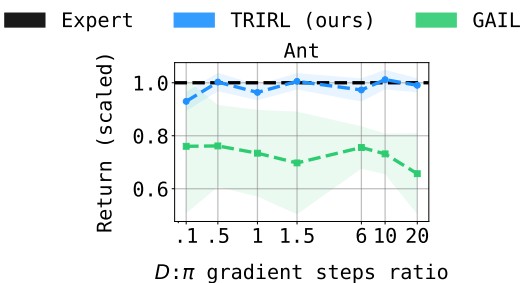

*Figure 7.* TRIRL is stable and highly performant across a wide range of hyperparameter values. Here, we show that TRIRL has stable performance, even with a near-perfect discriminator. In contrast, adversarial methods like GAIL are highly sensitive and typically fail due to the sharp decision boundaries induced by perfect discrimination.

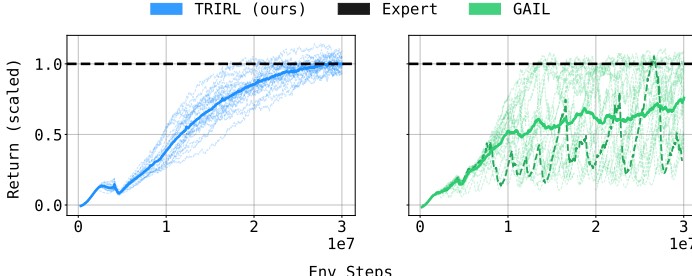

*Figure 8.* To underscore the monotonic performance improvement offered by our method, we plot all seeds from the Ant imitation learning experiment. TRIRL has much more stable and consistent training, and its performance grows approximately monotonically. In contrast, owing to its local rewards and suboptimal policies, GAIL arbitrarily fluctuates in performance during training (example seed in dark green).

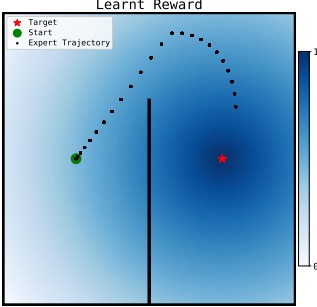

*Figure 9.* We show global reward functions learnt using a feature-based variant of our method, where we first learn a non-linear transformation of known base-features, and then a global reward function in this feature-space by employing a linear discriminator. We note this reward function is also transferable owing to it's general depiction of the desired "goal".

---

**Algorithm 2** Trust Region Inverse RL (continuous state-action spaces)

---

1: **Initialize:** neural networks $\pi^{(0)}$, $D^{(0)}$, and $R_{\text{fit}}^{(0)}$ ;
2: **Initialize:** circular buffer $\mathcal{B} = \{(D^{(i)}, \pi^{(i)}, R_{\text{fit}}^{(i)}, \eta^{(i)})\}_{i=0}^{k}$ ;
3: **Initialize:** trust-region bounds $\zeta_\mu, \zeta_\Sigma$ , $\eta_{\text{init}}$ , $\epsilon$ ;
4: **Output:** $r^\star$ and $\pi^\star$ ;
5: **repeat**
6:                                                     $\triangleright$ ROLLOUT
7:     $\{(\mathbf{s}_t, \mathbf{a}_t, \mathbf{s}'_{t+1})\}_{t=0}^{\text{rollout horizon}} \sim \text{rollout}(\pi^{(i)})$ ;
8:     learn $D^{(i)} \approx \log\left(\frac{\rho_E}{\rho_{\pi^{(i)}}}\right)$ by minimizing BCE loss ;
9:     append $D^{(i)}$ to $\mathcal{B}$ ;
10:                                         $\triangleright$ RETROSPECTIVE $\boldsymbol{\eta}$
11:     **if** retr. $\eta$ **then**
12:         $\{\eta^{(i)}\}_{i=0}^{k} \leftarrow \text{TR projection}\big(\{\pi^{(i)}\}_{i=0}^{k} \sim \mathcal{B}\big)$ ;
13:     **else**
14:         $\{\eta^{(i)}\}_{i=0}^{k} \sim \mathcal{B}$ ;
15:     **end if**
16:                                         $\triangleright$ REWARD CORRECTION
17:     $\{D^{(i)}\}_{i=0}^{k} \sim \mathcal{B}$ ;
18:     logits $\leftarrow \text{jax.vmap}\big(D^{(i)}.\text{predict}\big(\{(\mathbf{s}_t, \mathbf{a}_t, \mathbf{s}'_{t+1})\}\big)\big)_{\text{over } i}$ ;
19:     corrected reward $\leftarrow R_{\text{fit}}^{(i-k)}$ ;
20:     **for** logit$^{(i)}$, $\eta^{(i)}$ in zip(logits, $\{\eta^{(i)}\}_{i=0}^{k}$) **do**
21:         step $\leftarrow \epsilon/(1.0 + \eta^{(i)})$ ;
22:         corrected reward $\leftarrow (1.0 - \text{step}) \cdot \text{corrected reward} + \text{step} \cdot \beta \cdot \text{logit}^{(i)}$ ;
23:     **end for**
24:     intermediate reward $\leftarrow (1.0 - \epsilon) \cdot \text{corrected reward} + \epsilon \cdot \beta \cdot \text{logit}^{(i=k)}$ ;
25:     learn $R_{\text{fit}}^{(i+1)} \approx$ corrected reward ;
26:                                          $\triangleright$ POLICY OPTIMIZATION
27:     **if** TR loss **then**
28:         optimize $\pi^{(i+1)}$ to maximize intermediate reward $+ \text{schedule}(\eta_{\text{init}}) \cdot \text{TR loss}$ ;
29:     **else**
30:         $\eta^{(i+1)} \sim$ optimize $\pi^{(i+1)}$ to maximize intermediate reward with TRPL ;
31:     **end if**
32:     append $\eta^{(i+1)}$, $\pi^{(i+1)}$, and $R_{\text{fit}}^{(i+1)}$ to $\mathcal{B}$ ;
33: **until** converged

---

## B.1. Ablation Studies

Figure 10 shows how TRIRL scales with varying amounts of expert demonstrations. In this regard, our method is roughly comparable to the prior works considered in this paper, with the exception of NEAR, which performs much worse in low-data settings (because of the challenges of learning an accurate energy function). Figure 11 and Table 4 present an ablation study comparing the performance, runtime, and memory usage of TRIRL with different values of buffer size (k). We find that while a larger buffer size helps improve stability and performance, our method can also perform quite well with small values of buffer size, often outperforming baselines. We also find that the low-k configurations have much lower runtime and memory usage. Finally Tables 5 and 6 show ablation studies for the parameters $\beta$ and $\epsilon$.

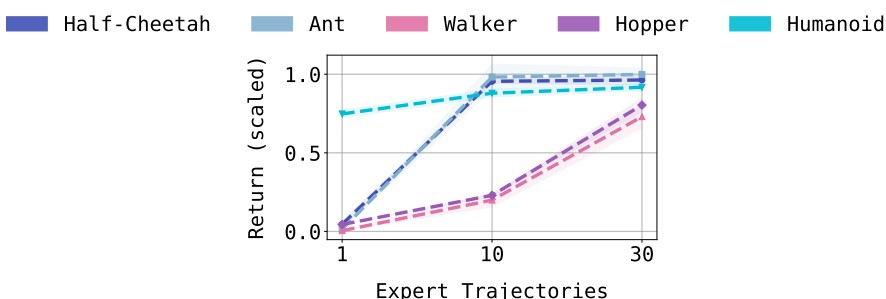

*Figure 10.* A plot showing how TRIRL scales with different amounts of expert demonstration trajectories.

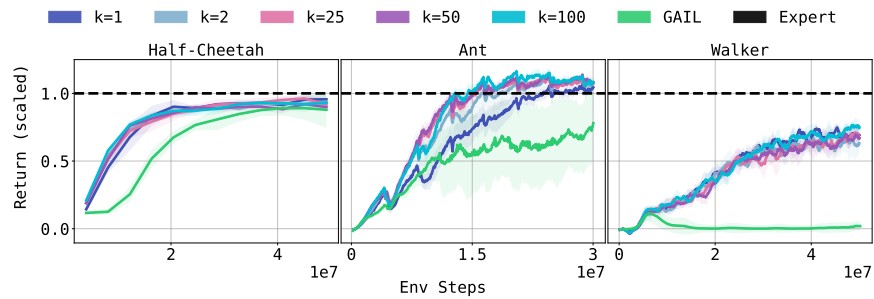

*Figure 11.* The scaling of performance with buffer size (k). TRIRL outperforms baselines even with very low values of buffer size (k). While a low k still beats baselines, performance and training stability benefit from larger k. This is expected since the contribution of reward fitting (and approximation errors) diminishes exponentially with k.

*Table 4.* The scaling of VRAM usage and runtime with buffer size (k) in the Mujoco Ant environment. TRIRL outperforms baselines even with very low values of buffer size (k). The low-k configurations have a negligible runtime and VRAM overhead.

| Algorithm | Memory (GB) | Runtime (sec/M steps) |
|---|---|---|
| TRIRL k=1 | **0.92** | **70.72 ± 0.96** |
| k=2 | 1.46 | 71.11 ± 0.98 |
| k=25 | 2.53 | 76.46 ± 1.32 |
| k=50 | 2.53 | 78.07 ± 1.11 |
| k=100 | 2.53 | 81.98 ± 3.33 |
| GAIL | 1.12 | 51.37 ± 2.13 |
| AMP | 3.69 | 61.76 ± 0.93 |
| AIRL | 1.27 | 66.51 ± 8.85 |
| NEAR | 1.01 | 100.41 ± 3.87 |
| LSIQ | 2.54 | 143.25 ± 12.86 |

*Table 5.* We conduct an ablation study to investigate the impact of the hyperparameter $\beta$ (entropy regularization). We find that TRIRL is generally insensitive to $\beta$ and the optimal entropy coefficient is generally task-dependent. Our results align with (Orsini et al., 2021, Fig.19).

| ent coeff | $\beta(1/\text{ent coeff})$ | Ant | Half Cheetah | Walker |
|---|---|---|---|---|
| 0.1 | 10 | $\mathbf{1.01 \pm 0.02}$ | $0.92 \pm 0.09$ | $0.67 \pm 0.08$ |
| 0.01 | 100 | $0.95 \pm 0.06$ | $\mathbf{0.95 \pm 0.09}$ | $\mathbf{0.73 \pm 0.08}$ |
| 0.001 | 1000 | $1.00 \pm 0.08$ | $0.91 \pm 0.10$ | $0.67 \pm 0.08$ |
| 0.0001 | 10000 | $1.00 \pm 0.06$ | $0.93 \pm 0.08$ | $0.67 \pm 0.09$ |
| 0.00001 | 100000 | $0.94 \pm 0.12$ | $0.92 \pm 0.08$ | $0.73 \pm 0.08$ |

*Table 6.* We conduct an ablation study to investigate the impact of the hyperparameter $\epsilon$ which controls the ratio of newly fitted log density ratios and the corrected reward from the previous iteration. TRIRL is also not overly sensitive to $\epsilon$. We find that, setting this to a low value generally ensures good performance, though higher values can speed-up convergence at the cost of performance guarantees.

| $\epsilon$ | Ant | Half Cheetah | Walker |
|---|---|---|---|
| 0.01 | $\mathbf{1.02 \pm 0.03}$ | $0.94 \pm 0.08$ | $0.75 \pm 0.03$ |
| 0.2 | $0.99 \pm 0.05$ | $\mathbf{0.95 \pm 0.09}$ | $\mathbf{0.76 \pm 0.03}$ |
| 0.4 | $1.01 \pm 0.03$ | $0.91 \pm 0.08$ | $0.75 \pm 0.06$ |
| 0.6 | $1.00 \pm 0.06$ | $0.91 \pm 0.10$ | $0.72 \pm 0.09$ |
| 0.8 | $0.99 \pm 0.02$ | $0.91 \pm 0.08$ | $0.69 \pm 0.04$ |
| 0.99 | $0.95 \pm 0.12$ | $0.90 \pm 0.09$ | $0.68 \pm 0.10$ |

# C. Experiment Details

## C.1. Baselines

**Generative Adversarial Imitation Learning (GAIL):** Ho & Ermon (2016) propose GAIL, one of the first methods to reformulate the MCE-IRL dual ascent algorithm (Ziebart et al., 2010) into a scalable method for imitation learning in complex continuous-domain settings. GAIL achieves the same optimal primal solution as MCE-IRL, however, it relies on per-step local policy optimization and a local reward function learnt using a neural network discriminator. GAIL minimizes the Jensen-Shannon divergence between the expert and agent distributions. Practically, the reward function is defined as $\log \sigma \left( D(\mathbf{s}, \mathbf{a}) \right)$ where $D(\mathbf{s}, \mathbf{a})$ is a binary classifier that distinguishes expert and agent samples. While the paper originally proposes TRPO (Schulman et al., 2015) for policy optimization, most modern implementations use PPO (Schulman et al., 2017) because of its better performance and scalability.

**Adversarial Inverse Reinforcement Learning (AIRL):** Fu et al. (2018) present AIRL, another adversarial IL algorithm that learns unshaped rewards and focuses on reward recovery. Similarly to our method, AIRL minimizes the reverse KL divergence between the expert and agent samples. The main contribution of the paper is to introduce an unshaped reward function to disentangle the reward from the environment's dynamics. In doing so, Fu et al. (2018) claim to learn a reward function that conveys the expert's motivations, instead of simply rewarding closeness to the expert's transitions. In AIRL, the discriminator takes the form:

$$D(\mathbf{s}, \mathbf{a}) = \frac{\exp \left( f_{\theta, \phi}(\mathbf{s}, \mathbf{a}) \right)}{\exp \left( f_{\theta, \phi}(\mathbf{s}, \mathbf{a}) \right) + \pi(\mathbf{a}|\mathbf{s})} \quad \text{where}$$

$$f_{\theta, \phi}(\mathbf{s}, \mathbf{a}) = g_\theta(\mathbf{s}, \mathbf{a}) + \gamma h_\phi(\mathbf{s}') - h_\phi(\mathbf{s})$$

Here, the unshaped rewards are learnt as $g_\theta(\mathbf{s}, \mathbf{a})$ or $g_\theta(\mathbf{s})$ (in case of state-only rewards). Similarly to GAIL, the authors propose TRPO for policy optimization, however, modern implementations use PPO.

**Adversarial Motion Priors (AMP):** Recently, Peng et al. (2021) propose AMP, a method that leverages the improvements from least-squares GANs (Mao et al., 2017), for adversarial imitation learning. Mainly, AMP differs from GAIL in its minimization of the $\chi^2$ divergence between the expert and agent distributions. Peng et al. (2021) also engineer the reward function slightly differently to better leverage the squared-error minimization. The AMP reward function is $r(\mathbf{s}, \mathbf{a}) = \max[0, 1 - 0.25(D(\mathbf{s}, \mathbf{a}) - 1)^2]$ where $D(\mathbf{s}, \mathbf{a})$ is a binary classifier trained to assign a label of 1 to the expert samples and -1 to the agent samples.

**Least Squares Inverse Q-Learning (LSIQ):** Garg et al. (2021) introduce Inverse Soft Q-Learning (IQ-Learn), a non-adversarial method that reformulates MCE-IRL in the Q-policy space. IQ-Learn uses the inverse soft Bellman operator $\mathcal{T}^\pi \mathcal{Q}(\mathbf{s}, \mathbf{a}) = Q(\mathbf{s}, \mathbf{a}) - \gamma \mathbb{E}_{\mathbf{s}' \sim \mathcal{P}}[V_\pi(\mathbf{s}')]$ to reformulate the entropy regularized distribution matching objective into an objective that only depends on the Q function. They do this by leveraging the fact that the optimal policy depends on $Q(\mathbf{s}, \mathbf{a})$ in closed form. Ultimately, the Q function is learnt by minimizing

$$\mathcal{J}(\pi, Q) = \mathbb{E}_{\rho_E}\left[\phi\left(Q(\mathbf{s}, \mathbf{a}) - \gamma \mathbb{E}_{\mathbf{s}' \sim \mathcal{P}}[V_\pi(\mathbf{s}')]\right)\right] - (1 - \gamma)\mathbb{E}_{\rho_{\not\pi}}[V^\pi(\mathbf{s}_0)]$$

and the policy is learnt using SAC (Haarnoja et al., 2018). Al-Hafez et al. (2023a) extend IQ-Learn by leveraging a mixture distribution for computing the expectation in $\mathcal{J}(\pi, Q)$. The resulting optimization problem is shown to minimize a bounded $\chi^2$ divergence between the expert and the mixture distribution. The resulting method, called LSIQ, also properly handles absorbing states (discussed further in Appendix C.4), leading to a better performing algorithm. In our motion-capture imitation experiments on the G1 humanoid robot, we use the state-only $Q$-function objective from (Garg et al., 2021, Appendix A.1) within LSIQ.

**Noise Conditioned Energy Based Annealed Rewards (NEAR):** Diwan et al. (2025) present NEAR, an energy-based imitation learning method that uses score-based models to learn a smooth, stationary reward function that can then directly be optimized by reinforcement learning. NEAR uses noise-conditioned score networks (Song & Ermon, 2019), a score-based generative model, to approximate the expert distribution as an unnormalized energy function $E_\theta(\mathbf{s}, \mathbf{a}, \sigma)$. Given a perturbation level defined by variance $\sigma^2$, the energy function is shown to directly correspond to a reward function. Diwan et al. (2025) then show that this energy function can be optimized with PPO to learn imitation policies. The paper also introduces an annealing framework to gradually change the $\sigma$ curriculum to improve performance. One of the primary drawbacks of NEAR is that learning an expressive energy-based reward function requires a lot more expert demonstrations that most traditional IRL methods. Hence, NEAR often underperforms in our low-data experimental settings (especially on the G1 and Go2 robots).

**Successor Feature Matching (SFM):** Jain et al. (2025) introduce SFM, a non-adversarial IL method that directly matches the expert and agent's successor features instead of learning an explicit discriminator or reward function. SFM assumes a base feature map $\phi(\mathbf{s})$ and represents long-horizon feature occupancy through successor features $\psi^\pi(\mathbf{s}, \mathbf{a}) = \mathbb{E}_\pi[\sum_{t=0}^\infty \gamma^t \phi(\mathbf{s}_t) \mid \mathbf{s}_0 = \mathbf{s}, \mathbf{a}_0 = \mathbf{a}]$. The main observation is that, under a linear reward class, the witness reward that maximally distinguishes the expert and agent can be written in closed form as the difference between their expected successor features. Concretely, SFM estimates $\hat{\mathbf{w}} = \hat{\psi}^E - \hat{\psi}^\pi$ and uses the induced reward direction $r(\mathbf{s}) \propto \phi(\mathbf{s})^\top \hat{\mathbf{w}}$ to update the policy. Rather than training a separate reward model, SFM uses the learnt successor features directly in the actor update, making the method a direct policy-search procedure for minimizing the feature-matching imitation gap. In practice, Jain et al. (2025) learn $\phi$ jointly with the policy and implement policy optimization with TD3/TD7-style actor-critic updates. The main drawback of SFM is that its performance depends on the quality of the learnt base features, and it offers no method for extracting the learnt reward function.

### C.2. Implementation Details

We conduct experiments on Mujoco continuous control benchmarks [2] as well as more challenging robotics tasks. The Mujoco environments are described in detail on the official website linked below. For the robotics experiments, we use the Unitree G1 and Unitree Go2 robots. The G1 is a 35 kg, 1.3 m tall humanoid robot capable of dynamic locomotion tasks. It is a 23 degree-of-freedom system with high-torque quasi-direct-drive actuators for greater speed and precise control. The G1 simulation environment has a 256-dimensional observation space composed of base angular velocity and roll/pitch, full joint positions and velocities, the previous action, and a short history stack of recent proprioceptive states. The G1 has a 23-dimensional action space, where each component is a target joint position increment passed through a PD controller (position control). The Go2 is a 15.8 kg, 0.67 m long quadruped robot designed for dynamic legged locomotion. It is a 12 degree-of-freedom system with actuated joints in each of its four legs. The Go2 simulation environment has a 42-dimensional observation space composed of trunk linear and angular velocities, full joint positions and velocities, and the previous action. It has a 12-dimensional action space, where each component is a target joint position passed through a PD controller. Figure 12 shows a snapshot of all environments.

We leverage massively parallel reinforcement learning environments for all our experiments (Bohlinger & Dorer, 2023). To

---

[2]https://gymnasium.farama.org/environments/mujoco/

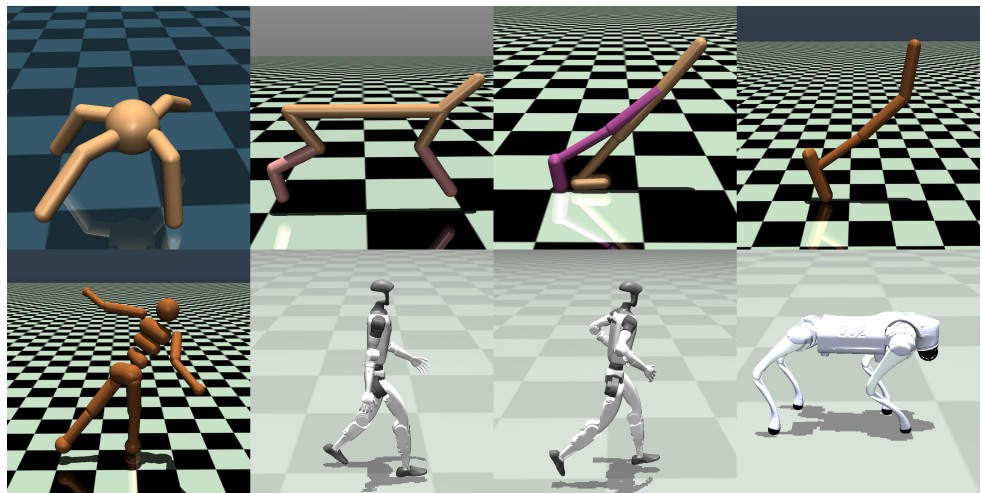

*Figure 12.* Environments used in our experiments.

this end, we use the Mujoco XLA simulation framework (Todorov et al., 2012) to handle all algorithm and environment computations on the GPU. Further, for a fair comparison, we implement all methods using the just-in-time compiled Python frameworks, Jax/Flax (Bradbury et al., 2022; Heek et al., 2024). This allows us to parallelize multiple algorithmic components for all methods efficiently. Such parallelization is especially useful for quickly computing the several intermediate quantities described in Sections 4.2 and 4.3. Specifically, we rely on parallelization for:

- *Trust-region projections:* computing Lagrangian multipliers for all samples in parallel.

- *Reward interpolations:* running inference on all discriminators in the buffer at once.

- *Retrospective Lagrangian multipliers:* computing Lagrangian multipliers for all samples across all past policies in the buffer.

### C.3. Trust Region Projections

Here, we briefly elaborate on the algorithmic and implementation details of computing trust region policy projections (Section 4.3). Given a Gaussian policy $\pi(\cdot|\mathbf{s}) = \mathcal{N}(\mu_\theta(\mathbf{s}), \Sigma_\phi)$, Otto et al. (2021) propose a method to exactly enforce the analytical reverse KL trust region by the application of a projection layer during policy optimization. The projection layer maps every violating policy prediction back into the trust region such that the projected policy parameters $(\tilde{\mu}_\mathbf{s}, \tilde{\Sigma})$ solve the following optimization problem:

$$\arg\min_{\tilde{\mu}_\mathbf{s}} d_{\text{mean}}\left(\tilde{\mu}_\mathbf{s}, \mu_\theta(\mathbf{s})\right), \quad \text{s.t.} \quad d_{\text{mean}}\left(\tilde{\mu}_\mathbf{s}, \mu_{\text{old}}(\mathbf{s})\right) \leq \zeta_\mu,$$

$$\arg\min_{\tilde{\Sigma}} d_{\text{cov}}\left(\tilde{\Sigma}, \Sigma_\phi\right), \quad \text{s.t.} \quad d_{\text{cov}}\left(\tilde{\Sigma}, \Sigma_{\text{old}}\right) \leq \zeta_\Sigma, \tag{6}$$

where, $d_{\text{mean}} = \text{\textonehalf}\left((\mu_2 - \mu_1)^T \Sigma_2^{-1}(\mu_2 - \mu_1)\right)$ and $d_{\text{cov}} = \text{\textonehalf}\left(\log |\Sigma_2|/|\Sigma_1| + \text{tr}\{\Sigma_2^{-1}\Sigma_1\} - d\right)$ are the mean and covariance components of the reverse KL divergence for two Gaussian distributions with means $\mu_1$ and $\mu_2$, covariances $\Sigma_1$ and $\Sigma_2$, and dimensionality $d$. The projected policy's mean is known in closed form as:

$$\tilde{\mu}_s = \frac{\mu_\theta(\mathbf{s}) + \eta_\mu(\mathbf{s})\,\mu_{\text{old}}(\mathbf{s})}{1 + \eta_\mu(\mathbf{s})} \quad \text{with Lagrangian multiplier}$$

$$\eta_\mu(\mathbf{s}) = \sqrt{\frac{(\mu_{\text{old}}(\mathbf{s}) - \mu_\theta(\mathbf{s}))^T \Sigma_{\text{old}}^{-1}(\mu_{\text{old}}(\mathbf{s}) - \mu_\theta(\mathbf{s}))}{\zeta_\mu}} - 1$$

For the covariance, Otto et al. (2021) compute the projected policy's precision $\tilde{\Lambda} = \tilde{\Sigma}^{-1}$ as an interpolation between precision matrices:

$$\tilde{\Lambda} = \frac{\eta_\Sigma^\star \, \Lambda_{\text{old}} + \Lambda}{\eta_\Sigma^\star + 1}, \quad \eta_\Sigma^\star = \arg\min_{\eta_\Sigma} \, g(\eta_\Sigma), \text{ s.t. } \eta_\Sigma \geq 0$$

where $\eta_\Sigma$ is the covariance Lagrangian multiplier and $g(\eta_\Sigma)$ the dual function of Equation (6). This dual minimization cannot be solved in closed form, however Otto et al. (2021) formulate a differentiable trust region layer by solving the minimization using L-BFGS (Liu & Nocedal, 1989) and computing its gradients by taking the differentials of the KKT conditions of the dual. Following (Otto et al., 2021, Appendix B.5), we re-implement the trust region projection layer in Jax, using Blondel et al. (2022) for L-BFGS and `jax.custom_vjp` to compute its gradients. Crucially, Jax's parallelization allows us to compute these multipliers very quickly, even for very large rollout batches (approx. 41k samples at once).

### C.4. Absorbing State Handling

In traditional reinforcement learning settings, when an agent enters an absorbing state $\mathbf{s}_a$, it receives zero reward (i.e. $r(\mathbf{s}_a, .) = 0$), and the next state for any next agent action is always $\mathbf{s}_a$ (i.e. $\mathcal{P}(\mathbf{s}_a | \mathbf{s}_a, .) = 1$). This elegantly handles episode termination without hindering the RL objective of maximizing a known reward function. Kostrikov et al. (2019) highlight that this assumption, however, causes problems in inverse reinforcement learning — where the reward is learnt and the goal is to instead learn policies that imitate the expert. Primarily, the issue is that the optimal learnt reward can vary in scale arbitrarily, meaning that absorbing states are either seen are highly rewarding or highly costly. Depending on the task, this induces a survival/termination bias in the agent. Improper handling of absorbing states may cause some methods to over/under-perform relative to their true potential. As shown empirically in Kostrikov et al. (2019), adversarial methods like GAIL and AIRL (Ho & Ermon, 2016; Fu et al., 2018) can get a large performance gain by such biases. Al-Hafez et al. (2023a) also highlight that the same is true for non-adversarial methods like IQ-Learn (Garg et al., 2021).

For a fair comparison, we learn the reward function for absorbing states in all methods shown in this paper. For adversarial methods like GAIL, AIRL, and AMP (Peng et al., 2021), as well as our proposed algorithm (TRIRL), we handle absorbing states similarly to Kostrikov et al. (2019). i.e., we fit the discriminator on expert and agent samples in absorbing states, add an indicator variable in the discriminator input to indicate the absorbing state, and use the absorbing state value during advantage estimation:

$$V_\pi(\mathbf{s}) = \mathbb{E}_\pi \left[ r(\mathbf{s}, \mathbf{a}) + (1 - \nu) \, \gamma V_\pi(\mathbf{s}') + \nu \, V(\mathbf{s}_a) \right] \quad \text{where}$$
$$V(\mathbf{s}_a) = \frac{\gamma}{1 - \gamma} \, r(\mathbf{s}_a) \quad ; \quad r(\mathbf{s}_a) \text{ is learnt} \quad ; \quad \nu = \mathbf{1}_{\{\mathbf{s}' \text{ is absorbing}\}}.$$

For NEAR (Diwan et al., 2025), we train the energy-based reward on absorbing states and follow the same procedure for advantage estimation. LSIQ (Al-Hafez et al., 2023a) rectifies absorbing state handling in IQ-Learn, and we implement it as described in the paper.

### C.5. Hyperparameters

The following hyperparameters were fixed across all methods in this paper. For the policy, we use a 3-layer dense network and predict the `log` standard deviation to ensure non-negativity. For the simpler, Mujoco tasks we use 256 neurons per layer and for the harder, robotics tasks we use [512, 256, 128] neurons respectively. We use `tanh` activations between all layers and clip actions to the environment's action range. We use the same architecture for the discriminator and critic networks (except for the robotics tasks where we use [1024, 512] neurons for the critic). We use a rollout horizon of 10 steps and set the following RL hyperparams: { $\gamma : 0.99$, $\lambda_{\text{gae}} : 0.95$, PPO clipping: 0.2, gradient clipping: 1.0, gradient penalty: 0.005, mini-batch size: 512 }. We tune learning rates and the number of gradient updates for each algorithm. In general, we found optimal values in the following ranges: { $\text{LR}_\pi : 4e^{-5}$ - $1e^{-4}$ , $\text{LR}_{\text{disc}} : 8e^{-5}$ - $3e^{-4}$, grad steps: 20 - 30}. For adversarial methods, we found that significantly fewer discriminator gradient steps were needed (approx. 3 - 10). We also tune the entropy coefficient ($1/\beta$ in our case) specifically for each environment. In general, we found that values in the range $5e^{-3}$ - $8e^{-5}$ perform the best. Table 7 shows hyperparameters specific to our method. Table 8 reports raw expert and random returns.

*Table 7.* Hyperparameters specific to our method.

| Hyperparameter | Optimal Range | Tuning Recommendation |
|---|---|---|
| $\epsilon$ | 0.2 - 0.4 | start low & increase for faster convergence |
| disc. buffer capacity | 80 - 120 | start high & decrease for faster runtime |
| **max $\eta$** | | |
|    mean bound | 0.0002 - 0.005 | refer to Otto et al. (2021) |
|    cov bound | 0.0001 - 0.004 | " |
|    TR regression loss coef. (Otto et al., 2021, Section 4.4) | 0.4 - 0.6 | " |
| **TR loss** | | |
|    $\eta_{\text{init}}$ | 60 - 100 | start high & decrease for faster convergence |

*Table 8.* Expert and random policy performance. [†] The G1 experiments use motion-capture data for the expert dataset. For this task, we normalize with respect to a "perfect imitation" score with the per step score as $\exp(-(v_\pi - v_{\pi_E})^2)$, where $v_\pi$ and $v_{\pi_E}$ are the agent's and expert's upper-body velocities.

| Environment | Expert Performance | Random Performance |
|---|---|---|
| Half Cheetah | 5050.39 | -310.55 |
| Half Cheetah (wind) | 4841.05 | -369.77 |
| Half Cheetah (mars grav.) | 4100.14 | -418.77 |
| Ant | 5390.62 | 94.43 |
| Ant-Disabled | 4377.28 | 76.59 |
| Walker | 5480.83 | 37.53 |
| Hopper | 3531.76 | 24.56 |
| Humanoid | 7404.87 | 236.45 |
| Go2 | 2341.38 | 1.37 |
| G1[†] | 1000.00 | 2.86 |

**Notes on SAC Training:**    As mentioned above, we leverage massively parallel simulators in our reinforcement learning setup. These simulators greatly reduce training time and make better use of the available computational resources. In our analysis, all baselines except LSIQ (Al-Hafez et al., 2023a) use PPO (Schulman et al., 2017) for policy optimization. Being on-policy, PPO scales very easily with parallelized environments and allows us to greatly scale up training speed for all methods. LSIQ, however, relies on SAC (Haarnoja et al., 2018) for policy optimization in continuous settings. This is a drawback because SAC is difficult to scale with parallel simulations. Without parallelized environments, we found that LSIQ (as well as IQ-Learn (Garg et al., 2021)) takes roughly 12–18 hours (about 8-10x the mean runtime of the other methods) to reach the same number of training samples as the baselines, while performing roughly equally. To ensure a fair comparison with a similar computational budget for all methods, we tune LSIQ specifically for parallelized environments. In this context, Li et al. (2023) identify key challenges in scaling SAC and provide hyperparameter recommendations for improving training performance. Their main suggestions are to greatly increase the replay buffer capacity, the batch size, and the critic–policy update ratio. Following these recommendations, we tuned LSIQ to perform well in parallelized simulators while requiring much less runtime than the non-parallelized version. Because of these scaling issues, we also considered the number of environments as a hyperparameter for LSIQ tuning. We also note that both the parallel and single-environment versions of LSIQ reached similar maximum performance when using a tuned set of hyperparameters. Sample efficiency could be further improved by considering recent advancements in using batch/weight norms in SAC (Palenicek et al., 2026).

