# OpenReview forum: "Trust Region Inverse Reinforcement Learning: Explicit Dual Ascent using Local Policy Updates"
_ICML.cc/2026/Conference — ICML 2026 regular_

### Official Review · Reviewer_P7D4 · 2026-03-12

**Soundness:** 3
**Presentation:** 3
**Significance:** 3
**Originality:** 3
**Overall Recommendation:** 5
**Confidence:** 3

**Summary:**

This paper proposes TRIRL: it first performs a reward update in function space, then executes a trust-region policy update restricted to a neighborhood of the previous policy, and subsequently applies a reward correction using the corresponding Lagrange multiplier. The method aims to preserve the monotonic dual improvement structure of classical MCE-IRL without requiring a full RL solve at every iteration, while learning a reward function that can be re-optimized from scratch and transfers, to some extent, to settings with changed system dynamics. Compared to adversarial imitation learning methods, TRIRL also demonstrates clear advantages in training stability.

**Compliance With Llm Reviewing Policy:**

Affirmed.

**Final Justification:**

Thanks for the authors' responses, my concerns have been resolved.

**Key Questions For Authors:**

**Q1.** Could the authors evaluate reward transferability on tasks beyond Ant-Disabled, or under larger magnitudes of dynamics shift? (See W1)

**Q2.** Could the authors provide the ablation result described in Weakness 2, i.e., retaining both TRPL and the discriminator buffer while setting η = 0 to remove reward correction? (See W2)

**Q3.** In resource-constrained settings, what guidance can the authors offer for selecting the buffer size k, and how does this choice affect approximation quality and training stability? (See W3)

**Q4.** Could the authors provide practical guidance on how to set the hyperparameters when applying TRIRL to a new task? (See W4)

**Q5.** The results in Table 1 show a substantial performance drop for TRIRL state-only in both the re-training and transfer settings relative to training performance (0.83 → 0.22 and 0.20, respectively). How do the authors explain this large gap?

**Limitations:**

Yes.

**Strengths And Weaknesses:**

## Strengths

**Soundness:** The problem setting is clearly defined and the technical approach is largely self-consistent. The core theoretical results (Lemma 3.1 and Theorem 3.2) are accompanied by complete proofs, and the alignment between the reward update direction and the gradient of the dual objective is rigorously derived in Appendix A. The experimental evaluation covers a broad range of tasks with a reasonably complete design, and the authors discuss the limitations of their method with commendable honesty.

**Presentation:** The paper is well-structured and the theoretical development in the methods section is coherent and easy to follow.

**Significance:** This work addresses how to simultaneously preserve the classical IRL structure — interpretable, re-optimizable, and meaningful for reward recovery — while scaling to deep continuous control and robotics settings. Compared to methods that focus solely on improving imitation return, TRIRL additionally aims to recover a reward function that can be re-optimized from scratch and retains some effectiveness under system dynamics shifts. This is particularly valuable for applications where IRL is used for intent inference or reward learning rather than pure imitation.

**Originality:** This paper bridges trust-region policy optimization and imitation learning through a reward-correction perspective: a trust-region optimal policy for a large-step reward update can be reinterpreted as the MCE-optimal policy for a corrected reward computed with a smaller step size in the same direction. This result reconnects local policy search back to the dual-ascent framework of classical MCE-IRL, and constitutes the most original and central contribution of the paper.

## Weaknesses

**W1. The claim of learning a globally transferable reward function requires stronger empirical support.** Learning a globally transferable reward function is one of the central claims of this paper, yet the transfer experiments are conducted on only a single task (Ant-Disabled) with a single type of dynamics change. We recommend that the authors validate this claim more systematically across multiple tasks and a wider range of dynamics perturbations.

**W2. The ablation study is missing a critical control.** We recommend that the authors add an ablation that retains both TRPL and the
discriminator buffer while fixing η = 0, thereby removing the reward correction. Existing configuration (v) corresponds to "buffer + η = 0" without TRPL, and configuration (vi) corresponds to "TRPL" without the buffer; neither can serve as a substitute for this control. Without it, the paper cannot directly isolate and verify the independent contribution of reward correction as the core dual-correction mechanism.

**W3. The discussion of computational cost is insufficient.** The reported VRAM usage and wall-clock training times correspond to a fixed buffer size, but the paper does not discuss how resource consumption scales as k increases, despite k being a key parameter governing approximation quality.

**W4. The method involves a relatively large number of hyperparameters with wide optimal ranges.** TRIRL introduces a substantial number of method-specific hyperparameters (Table 3), which may make it difficult to transfer the method to new settings without extensive tuning.

---

> ### Author Rebuttal · Authors · 2026-03-31
>
> We thank the reviewer for their feedback and for suggesting insightful additional experiments and ablations. We conducted the suggested studies and have addresses all the questions below.
>
> **New Experiments (anonymous):** https://anonymous.4open.science/r/trirl_rebuttal-304E/
>
> Q1/W1: Please refer to our reply to reviewer CjCM. We validate the global/transferrable rewards claim systematically by improving our results and adding 3 more retraining/transfer tasks (Tables D1, D2).
>
> Q2/W2: Thank you for suggesting this crucial ablation. We agree that this control is important to justify the claim that TRIRL’s improvements are due to explicit dual optimization via its reward correction mechanism. We provide this ablation through FigB in our anonymous repository.
>
> Q3/Q4/W3: Thank you for pointing this out. We agree that buffer size (k) is a crucial hyperparameter and present some additional ablation experiments that help quantify its necessity and contribution to runtime/VRAM. The FigA and TableA show an ablation study comparing different values of buffer size (k) vs performance, runtime, and VRAM. We show that **TRIRL outperforms baselines even with very low values of  buffer size (k)**. The **low-k configurations have a negligible runtime and VRAM overhead** (compared to adversarial methods like GAIL/AIRL) while also outperforming baselines. Below, we provide some additional observations, implementation details, and tuning guidance.
> - While a low k still beats baselines, **performance and training stability benefit from larger k**. This is expected since the contribution of reward fitting (and approximation errors) diminishes exponentially with k.
> - **VRAM usage is constant after a certain value of k**. In our implementation, we leverage a combination of chunked inference and jit-compilation. The discriminator parameters are stored in CPU memory, and loaded in chunks of size $c < k$ at inference time. After inference, the parameters are offloaded immediately, which (i) significantly reduces our VRAM needs and (ii) allows for budget-based VRAM usage. We can apply our method in resource constrained settings with smaller c or leverage more VRAM with larger c. TableA shows a constant VRAM usage after k $\geq$ 25 (we chose c=20).
> - **Runtime is roughly comparable to baselines and scales well**.
> - **Tuning:** our core mechanism requires only two hyperparameters ($\epsilon$ and k). $\epsilon$ controls the influence of newly fitted log density ratios. For hard tasks with large observation spaces — where density ratio estimation is prone to be erroneous — we recommend starting with lower $\epsilon \in [0.1, 0.3]$. k controls how much the approximation errors of reward fitting affect our method. We recommend starting high ($\approx 100$) and decreasing based on resource availability. Trust region bounds can be set as per the guidelines in Otto et al. 2021. $\eta_{\text{init}}$ can be set conservatively ($\approx 80-100$) and reduced if convergence is slow.
>
> W4 _(also relevent for reviewer eVoq)_: We would like to clarify that not all hyperparameters in Table 3 are relevent for any single variant of our method. The core components of our method only need 2 hyperparameters ($\epsilon$ and buffer capacity k). The other hyperparameters are related to trust region policy optimization. The TR loss version just additionally needs an initial Lagrangian multiplier $\eta_{\text{init}}$. The versions based on TRPL require the TRPL-specific params (trust region bounds, and an additional optional amortization loss coefficient). We also ablate $\beta$ and $\epsilon$ and present the results in Tables B and C.
>
> **Hyperparameter Tuning** _(also relevent for reviewer eVoq)_: We tuned all important hyperparameters necessary for each algorithm (not just learning rates and gradient steps).  Each method was tuned using Bayesian optimization (using wandb.ai) till a single competitive set of hyperparameters was obtained. While we did not impose strict compute-threshold-based budgets, we also did not favour our method and gave each method roughly the same hyperparameter tuning budget. In fact, we spent considerable effort in tuning methods like AIRL and LSIQ on the harder tasks to really compare our method against the most competitive versions of these baselines.
>
> Following Orsini et al. (2021), we tuned the main method-specific parameters and kept the core RL hyperparameters fixed across methods for fairness. We tested both human and synthetic data; auxiliary settings were not tuned.
>
> Q5: Please refer to our reply to reviewer CjCM. In addition to new experiments, we also analyse specific reasons for poor retraining and transfer performance, then improve our results using a simple change. We also show the learnt global reward function in the point maze task in FigC.
>
> ---
>
> Orsini M et al. What matters for adversarial imitation learning?. NeurIPS 2021.

---

> > ### Author Rebuttal · Reviewer_P7D4 · 2026-04-02
> >
> > Thanks for the authors' responses. Most of my concerns have been resolved. However, I am still curious about the cost of hyperparameter tuning, therefore my following-up question is:
> > Q1. The authors mentioned that "each method was tuned using Bayesian optimization (using wandb.ai) till a single competitive set of hyperparameters was obtained", could the authors provode the expected time required for the proposed method to search for the optimal parameters on a new dataset?

---

> > > ### Author Response · Authors · 2026-04-04
> > >
> > > We thank the reviewer for their continued engagement with our submission. Below, we give additional data on tuning. We show:
> > > - Wandb sweep statistics for the Ant and G1 Walk tasks (total runs in the sweep, when the best config was obtained, and the cumulative GPU time till that point). We also compare the tuning time to the achieved performance.  **In most cases, TRIRL showed the best tunability and had the best compute vs. performance tradeoff**.
> > > - The sweep ranges for parameters. Here [a,b] is a range and [a,b,c,d] is a set of available options.
> > >
> > > | Algorithm/Env   | Total Wandb Runs | Num. Runs Till Best | GPU Time Till Best (hours) | Tuned Performance |
> > > | ----------- | ---------------: | ------------------: | -------------------------: | ---------------------------: |
> > > | **Ant**     |                  |                     |                            |                              |
> > > | TRIRL       |               30 |                  14 |                      13.72 |                 0.99 +- 0.05 |
> > > | GAIL        |               50 |                  31 |                      13.27 |                 0.76 +- 0.31 |
> > > | AIRL        |               30 |                   3 |                       1.66 |                 0.45 +- 0.19 |
> > > | AMP         |               30 |                  26 |                      13.38 |                 0.67 +- 0.11 |
> > > | NEAR        |               30 |                   7 |                       5.86 |                 0.46 +- 0.31 |
> > > | LSIQ        |               50 |                  13 |                      15.52 |                 0.47 +- 0.24 |
> > > | **G1 Walk** |                  |                     |                            |                              |
> > > | TRIRL       |               30 |                   4 |                       8.80 |                 0.50 +- 0.27 |
> > > | GAIL        |               30 |                  30 |                      46.31 |                 0.22 +- 0.29 |
> > > | AIRL        |               60 |                  54 |                      48.92 |                 0.02 +- 0.00 |
> > > | AMP         |               30 |                  11 |                      10.51 |                 0.05 +- 0.16 |
> > > | NEAR        |               30 |                   - |                          - |                        |
> > > | LSIQ        |               30 |                   - |                          - |                        |
> > >
> > > **Sweep Ranges**
> > > ```bash
> > > # Adversarial IL & Common
> > > entropy_coef: [1e-5,1e-2]
> > > learning_rate: [1e-5,1e-3]
> > > learning_rate_disc: [1e-6,1e-3]
> > > nr_grad_steps: [5,10,20,30]
> > > nr_grad_steps_disc: [1,3,5,10,20,30]
> > >
> > > # NEAR
> > > learning_rate_ncsn: [1e-5,1e-3]
> > > L_ncsn: [20,50,80]
> > > batch_size_ncsn: [256,512,1024]
> > > minibatch_size_ncsn: [64,128,256]
> > > buffer_size: [1e6,3e6,5e6,8e6,1.2e7] #included larger buffers only in harder tasks
> > > nr_grad_steps_ncsn: [10,20,30]
> > >
> > > # LSIQ
> > > batch_size: [1024,2048,4096,8192,16384]
> > > minibatch_size: [128,256,512]
> > > nr_q_updates_per_step: [1,2,4,8,12,16]
> > > use_target_q: [true, false]
> > >
> > > # TRIRL (both TR loss & other variants)
> > > mean_bound: [1e-4,1e-2]
> > > cov_bound: [1e-4,8e-3]
> > > disc_buffer_capacity: [60, 120]
> > > epsilon: [0.1,0.9]
> > > init_eta: [60-120]
> > > learning_rate_reward_fn: [1e-5,1e-3]
> > > nr_grad_steps_rew: [10,20,30]
> > > trust_region_coef: [0.05,0.85]
> > > ```
> > >
> > > **Note:** We want to emphasize that this additional data is from our original sweeps for the parameters used in the paper. It is not meant as a properly designed experiment to compare the tunability of the different methods. It is somewhat affected by our experience with some baselines and the manual tuning needed for failed sweeps (blank lines in the table for LSIQ, AIRL, and NEAR in some cases). We present this data primarily for transparency. In general, based on our experience, TRIRL is comparably easy to tune,  which is consistent with the ablations that show low sensitivity with respect to the hyperparameters that are specific to our method.
> > >
> > > **We hope this additional data satisfies the reviewer’s questions and request that they consider revising their recommendation/confidence.**

---

### Official Review · Reviewer_tUJf · 2026-03-12

**Soundness:** 3
**Presentation:** 3
**Significance:** 3
**Originality:** 3
**Overall Recommendation:** 5
**Confidence:** 4

**Summary:**

Trust Region Inverse Reinforcement Learning (TRIRL) proposes a non-adversarial algorithm for inverse reinforcement learning that preserves monotonic improvement while avoiding the expensive inner-loop RL optimization required by classical Maximum Causal Entropy IRL. The method alternates between reward updates based on expert–agent occupancy mismatch and trust-region policy updates. A key theoretical result shows that a policy optimized within a KL trust region for an intermediate reward is globally optimal for a corrected reward with a smaller update step. This enables efficient dual optimization with local policy search. Experiments on MuJoCo and robotics tasks show improved stability, performance, and recovery of transferable

**Compliance With Llm Reviewing Policy:**

Affirmed.

**Key Questions For Authors:**

1.	How the trust region constraints are translated into the reward updates: \zeta in Equation (3) and \eta in Lemma 3.1?
2.	The operator in Lemma 3.1 is not explicitly defined. Can you define the operator explicitly?
3.	You keep the bounds on trust region fixed while the step size of reward update adjusted. How would you relate the changing step size to the bound on the trust region?

**Limitations:**

1.	Computational burden, esp. VRAM requirements
2.	Numerous parameters to tune: esp. Trust region bound
3.	Gaussian assumption required by TRPL

**Strengths And Weaknesses:**

Strengths:
•	TRIRL provides formal mathematical guarantees for monotonic improvement of the dual objective, matching the theoretical rigor of classical Maximum Causal Entropy IRL.
•	TRIRL achieves computational efficiency by replacing the exhaustive inner-loop reinforcement learning requirement with local trust-region updates.
•	The paper provides numerical studies using challenging high-dimensional environments such as  G1 and Go2.

Weaknesses:
•	TRIRL's implementation requires a circular buffer of historical discriminators to maintain recursive functional reward updates. This architectural choice imposes a substantial memory footprint, with VRAM requirements escalating from 1.1 GB in standard adversarial models to approximately 7.5 GB.
•	The framework relies on Differentiable Trust Region Projection Layers, which currently limit practical application to Gaussian policies. This modeling assumption is insufficient for complex tasks requiring multimodal action distributions, such as obstacle avoidance with multiple valid paths.
•	Trust-region optimization usually adaptively controls the bounds on trust-region depending on the improvement from the previous iteration, whereas the paper keeps the bounds fixed throughout.

---

> ### Author Rebuttal · Authors · 2026-03-31
>
> We wish to thank the reviewer for their comments and positive feedback. We believe we have been able to address all their points in the following paragraphs.
>
> **New Experiments (anonymous):** https://anonymous.4open.science/r/trirl_rebuttal-304E/
>
>
> > TRIRL's implementation requires a circular buffer of historical discriminators to maintain recursive functional reward updates. This architectural choice imposes a substantial memory footprint ...
>
> With regards to the discriminator buffer, we present some additional ablation experiments that help quantify its necessity and contribution to runtime/VRAM. We point the reviewer to our reply to reviewer P7D4. New results in TableA and FigA show an ablation study comparing different values of buffer size (k) vs performance, runtime, and VRAM. We show that TRIRL outperforms baselines even with very low values of  buffer size (k). The low-k configurations have a negligible runtime and VRAM overhead (comparable to adversarial methods like GAIL/AIRL) while also outperforming baselines. We further clarify that the 7.5 GB figure is an average over all tasks (which is skewed by harder robotics tasks), and that we can reduce the memory footprint by evaluating $c < k$ discriminators in chunks, at the cost of slightly increased computation time.
>
> > The framework relies on Differentiable Trust Region Projection Layers, which currently limit practical application to Gaussian policies...
>
> TRPL is not fundamentally necessary for our method: we point the reviewer to our reply to reviewer CjCM. We would like to clarify that our theory (including Theorem 3.2) itself is general and not restricted to Gaussian policies. In the continuous setting, we propose two variants of our method. One variant uses differentiable trust region projection using TRPL and is indeed limited to Gaussian policies. However, the other variant (called TR loss) is more general and is based on a trust region penalty (see $\mathcal{L}_{\text{tr}}$ in Section 4.3). The TR loss version does not make any assumptions on policy parameterization and also outperforms baselines in most tasks.
>
> ## Questions
>
> Q1: Here, we clarify how trust region constraints are used in TRIRL. As mentioned above, we propose two variants of our method:
> - In the “TR Loss”-variant, $\eta$ corresponds to the coefficient of the trust-region penalty ($\mathcal{L}_{\text{tr}}$ in Section 4.3) and is specified directly (the corresponding trust region $\zeta$ does not need to be known).
> - In the TRPL variant, we specify the trust-region constraint $\zeta$ and TRPL enforces these constraints and outputs $\eta$ as the Lagrangian multiplier.
>
> Q2: We believe there may be a minor typographical confusion in the question. Lemma 3.1 does not introduce new operators. However, Theorem 3.1 utilizes the reward update operator  $\mathcal{U}_{\epsilon}^{\rho}$ (defined in Eq. 2). This operator maps the current reward and occupancy mismatch to the updated reward function. We will ensure this is explicitly highlighted in the revision.
>
> Q3: Our derivations show that the reward update can account for a more local policy optimization (higher $\eta$) by using a smaller reward update $\epsilon\_\text{tr}$. In the TRPL variant we impose a trust-region constraint, and use the resulting Lagrangian multiplier $\eta$ to adapt the reward update $\epsilon\_\text{tr}$. In the TR-Loss variant, we decay the penalty term during optimization, which consequently will increase the reward step size $\epsilon\_\text{tr}$. In principle, we could also keep both factors constant, but unlike a fixed trust region, a fixed KL penalty typically doesn’t work well in RL, because the policy would either make too large steps in the early phase of the optimization, or too small steps in the late phase.
>
> ---
>
> Peters J, Mulling K, Altun Y. Relative entropy policy search. InProceedings of the AAAI Conference on Artificial Intelligence 2010 Jul 5 (Vol. 24, No. 1, pp. 1607-1612).
>
> Schulman J, Levine S, Abbeel P, Jordan M, Moritz P. Trust region policy optimization. InInternational conference on machine learning 2015 Jun 1 (pp. 1889-1897). PMLR.

---

> > ### Author Rebuttal · Reviewer_tUJf · 2026-04-05
> >
> > All of my questions have been answered and I do not have any more.

---

### Official Review · Reviewer_CjCM · 2026-03-13

**Soundness:** 3
**Presentation:** 3
**Significance:** 3
**Originality:** 3
**Overall Recommendation:** 5
**Confidence:** 4

**Summary:**

This paper proposes Trust Region Inverse Reinforcement Learning (**TRIRL**), a non-adversarial IRL framework that bridges the gap between mathematically rigorous but computationally expensive classical methods (e.g., MCE-IRL) and efficient but unstable adversarial methods (e.g., GAIL) . The key theoretical contribution is Theorem 3.2, which proves that a trust-region optimal policy for a given reward function can be viewed as a globally optimal policy for a corrected reward function with a modified step size. This allows the algorithm to perform stable dual-ascent updates with monotonic performance improvement guarantees without solving a full RL problem at each iteration. By leveraging this, TRIRL ensures monotonic performance improvement without requiring full RL optimization in the inner loop, enabling it to scale efficiently to high-dimensional tasks such as the 23-DOF Unitree G1 humanoid robot.

**Compliance With Llm Reviewing Policy:**

Affirmed.

**Final Justification:**

Most of my concerns have been addressed. I raise the score to 5 to reflect this.

**Key Questions For Authors:**

- Is there a clear path to extending the Theorem 3.2 mapping to non-Gaussian policy representations, such as Diffusion policies, GMMs, or programmatic RL policies [1] ?
- You claim TRIRL recovers a "global" reward function that captures the expert's intrinsic motivations. However, Table 1 shows a significant performance drop from 0.83 (during IRL training) to 0.22/0.32 during retraining from scratch. If the reward is truly global and sufficient to define the task, why is the performance of a newly initialized agent so much lower? Does this suggest that the recovered reward still lacks critical information or is partially entitled with the initial training distribution?

[1] Programmatic Reinforcement Learning without Oracles, ICLR 2022

**Limitations:**

Yes

**Strengths And Weaknesses:**

**Strengths**
- The paper establishes a bridge between cheap local policy updates and global reward optimality, inheriting the monotonic improvement guarantees of the MCE-IRL framework. This is important.
- By moving away from adversarial min-max games, TRIRL avoids common pitfalls like vanishing gradients and hyper-parameter sensitivity; its density ratio estimator remains stable even as the classifier approaches perfection.
- The experiments show TRIRL can successfully master a 256-dimensional observation space on the Unitree G1 humanoid. This proves the method's robustness for high-dimensional, real-world robotics.

**Weaknesses**
- The framework is currently restricted to Gaussian policies due to the requirements of the Trust Region Projection Layer (TRPL), potentially hindering the learning of multi-modal expert behaviors. I am curious whether this can be applied to other types of policies (see questions).
- The reward correction steps and the iterative numerical solver for TRPL introduce noticeable runtime overhead compared to simpler on-policy or adversarial alternatives.

---

> ### Author Rebuttal · Authors · 2026-03-31
>
> We thank the reviewer for the thorough review, positive comments, and questions. We address all weaknesses and questions below, and have provided new experiments to further support our claims.
>
> **New Experiments (anonymous):** https://anonymous.4open.science/r/trirl_rebuttal-304E/
>
> Q1/W1: We clarify that our theory (Theorem 3.2) is not limited to Gaussian policies but applies to any architecture and RL method capable of solving the MaxEnt-RL problem (since KL-penalized RL reduces to MaxEnt-RL). We propose two implementations: (i) TRIRL (projection layers, currently Gaussian) and (ii) the “TR-loss” variant (KL penalty, architecture-agnostic). Notably, both achieve strong performance. We used Gaussian policies as they are the standard, effective choice for the unimodal demonstrations in our benchmarks; multimodal likelihood-based policies (e.g., normalizing flows) are directly compatible with the “TR-loss”-variant. Policies without tractable probability, such as flow-matching or programmatic policies, are not fully consistent with our theory. Applying our framework to such cases, e.g. by employing lower-bound approximations (Celik et al., 2025), is a promising direction but outside our current scope. We will update our limitations to clarify that our variants require likelihood-based policies, with the TRPL-variant additionally assuming Gaussianity.
>
> W2: Reward correction itself does not contribute significantly to runtime, rather (i) the use of a discriminator buffer and (ii) the use of TRPL do. As highlighted before, TRPL is not fundamentally necessary for our method and we also propose an alternative (TR loss) version with much lower runtimes that often beats baselines. With regards to the discriminator buffer, we present some additional ablations that help quantify its necessity and contribution to runtime. We point the reviewer to our reply to reviewer P7D4 (weakness 3). New results in TableA and FigA show that **TRIRL outperforms baselines even with very low values of buffer size (k)**. The **low-k configurations have a negligible runtime and VRAM overhead** while also outperforming baselines.
>
> Q2: Thank you for raising this question. The ability to learn global rewards is a central claim and we now conducted additional experiments to support it. We will extend the discussion with a systematic analysis of the existing challenges. Namely, we identified two failure cases that limit the performance of the recovered reward function:
>
> - Practical implementations result in approximation errors; in particular, training a discriminator with binary cross entropy to approximate the log-density ratio typically results in large errors due to the non-linear optimization landscape and limited sample sizes.
> - The IRL problem is ill-posed due to temporal credit assignment. Different reward functions may result in the same optimal policy on a given environment, but to different behavior under different dynamics (Ng et al., 1999).
>
> The first issue is not a limitation of our theory, and indeed we demonstrate that we learn a global reward function in the toy task, where we do not suffer from such approximation errors, and are able to recover the expert policy also when retraining from scratch. The second issue only manifests in the transfer experiments (even on the toy task, as can be inferred from Fig.2). This ill-posedness is a well-known problem in IRL (Fu et al. 2018) and requires demonstrations under different dynamics or discount factors (Kim et al. 2021, Cao et al. 2021), which is a setting that we do not consider in the current work.
>
> However, we note that both issues can also be addressed by using inductive priors on the reward function, which we argue is often possible. For example, we could assume that the reward function for humanoid locomotion is a function of the floating-base velocity, torso height and action cost and aim to learn a nonlinear reward function of these features based on human demonstrations. Our IRL method does not require state-action observations and can therefore be used for learning such feature-based reward functions. Hence, we tested a variant for the rebuttal that first learns a non-linear extension of the base features (using a diffusion loss) to learn a suitable basis for approximating the expert densities, and then learns a linear reward function in those features by employing a linear discriminator. We used linear reward function for simplicity and interpretability, and as it enables us to drop the discriminator buffer. While this procedure is heuristic and only an early test of the potential of our method, **we obtain promising results both for retraining and transfer, that can be found in our anonymous repository**.
>
> ---
>
> Celik O, et al. DIME: Diffusion-Based Maximum Entropy Reinforcement Learning. ICML 2025
>
> Kim, K., et al.. Reward Identification in Inverse Reinforcement Learning. ICML 2021.Cao H., et al. Identifiability in inverse reinforcement learning. NeurIPS 2021.

---

> > ### Author Rebuttal · Reviewer_CjCM · 2026-04-04
> >
> > After carefully read eVoq's review, I think I need more input to judge the validity of the theoretical soundness of this work. I will keep my score to be a negative 3 temporarily.

---

> > > ### Author Response · Authors · 2026-04-04
> > >
> > > We thank the reviewer for their continued engagement with our submission. We believe that our method has a solid theoretical motivation, which is directly aligned with several strong and well-cited works in this field. Our rebuttal clearly pointed reviewer eVoq to these works, re-explained our method’s background, and situated our work in this field. However, these prior works have not been considered. Hence, we strongly disagree with  W1-3 raised by reviewer eVoq. Below, we give an extended response to these weaknesses and point the reviewer to fundamental literature in this field that unequivocally confirms our theoretical basis.
> > >
> > > W1: Reviewer eVoq argues that the regularizer changes the optimal solution, and strong regularization may result in almost uniform policies. We strongly disagree that this makes our method unsound. On the contrary, this is inherently how regularization works in machine learning, and the same argument could be made against L2-regularization in supervised learning, or against entropy regularization in MaxEnt-RL.
> > >
> > > We further emphasize that entropy regularization should indeed be considered as a regularizer in our setting, **not as an exploration bonus**. Entropy-regularized divergence minimization is very common in imitation learning and RL. While the original MCE-IRL formulation [1] does not trade off entropy maximization and expert-matching (using a hard constraint of matching expert expectations), [10 Section 5.1.4] already noted that relaxing this constraint can be useful in practice. While regularized MaxEnt-IRL methods traded off entropy regularization with minimizing absolute/squared distances in the feature expectations ($\ell_1$ or $\ell_2$ regularization on the reward params), our objective is based on the work by [2] who showed that entropy regularization combined with divergence minimization can be more effective than parameter-space regularization (c.f. Fig. 3). **Notably, entropy regularized divergence minimization is still today a very common objective in imitation learning. Eg:**
> > >
> > > - GAIL: $\min JS(p_\pi||p_E) - \alpha E[H(\pi)]$ (3, Eq. 15). We differ by the use of KL.
> > > - IQ-Learn discusses the generalized setting $\min_{\pi∈\Pi} d\psi (\rho_\pi, \rho_E) − H(\pi)$ (Garg et al. 2021, Eq. 4). Our objective is a special case obtained when using the scaled reverse KL for $d\psi$.
> > > - Other instances of similar objectives provided to reviewer eVoq [2, 4, 5, 9, 11].
> > >
> > > W2: The strong duality of the entropy regularised RL problem (Eq. (3)) has been studied deeply in prior work [6, 7, 8, 12]. As highlighted before, the entropy term is indeed concave. **[3, Lemma 3.1 & 12, Appendix A.1] show the concavity of the expected entropy w.r.t $\rho(s,a)$**, and that the expected entropy is the standard regularizer considered in IL/RL. We also provide a short derivation below:
> > >
> > > Eq (3) can be rewritten in occupancy variables $\rho(s,a)$ by expanding the expectation over $\rho(s)$. Given state marginal $\rho(s)=\sum_a \rho(s,a)$ and $\rho(s,a)=\rho(s)\pi(a\mid s)$, the outer expectation turns the KL constraint to be over state-action occupancy.
> > >
> > > $\text{Eq(3)} \equiv \max_{\rho}\; - \sum_{s,a}\rho(s,a)\log\frac{\rho(s,a)}{\rho(s)} + \sum_{s,a}\rho(s,a) r^{(i)}(s,a)$
> > >
> > > s.t.
> > > $\sum_{s,a}\rho(s,a)\log\frac{\rho(s,a)}{\rho(s)\pi^{(i)}(a\mid s)} \le \zeta, \quad
> > > \sum_a \rho^\pi(s',a)=\mu_0(s')+\gamma\sum_{s,a}P(s'\mid s,a)\rho^\pi(s,a)
> > > $
> > >
> > > The terms are concave (entropy) and linear (reward), with convex (KL) and linear (flow) constraints. Hence Slater’s conditions hold and strong duality applies. Finally, since $\rho(s,a)=\rho(s)\pi(a\mid s)$, the policy is equivalently obtained by optimizing over $\rho(s,a)$ followed by marginalization.
> > >
> > > W3: This has already been addressed in our initial rebuttal. We already include both projection (TRPL) and penalty-based variants in the paper.
> > >
> > > To conclude, we clarify that the theoretical concerns of reviewer eVoq have been fully addressed.
> > >
> > > ---
> > >
> > > [1] Ziebart et al. “Modeling interaction via the principle of maximum causal entropy”.
> > >
> > > [2] Arenz et al. “Optimal control and inverse optimal control by distribution matching”. IROS 2016
> > >
> > > [3] Ho et al. "Generative Adversarial Imitation Learning". NeurIPS 2016.
> > >
> > > [4] Osa et al. “An algorithmic perspective on imitation learning”. 2018
> > >
> > > [5] Ke L., et al. “Imitation learning as f-divergence minimization”. WAFR 2020
> > >
> > > [6] Peters J, et al. “Relative entropy policy search”. AAAI 2010
> > >
> > > [7] Achiam J, et al. Constrained policy optimization. ICML 2017
> > >
> > > [8] Geist M, et al. “A theory of regularized markov decision processes”. ICML 2019
> > >
> > > [9] Fu et al. “Learning robust rewards with adversarial inverse reinforcement learning”. ICLR 2018
> > >
> > > [10] Ziebart. “Modeling purposeful adaptive behavior with the principle of maximum causal entropy”. CMU. 2010.
> > >
> > > [11] Boularias et al. “Relative entropy inverse reinforcement learning”. AISTATS 2011
> > >
> > > [12] Neu at al. “A unified view of entropy-regularized markov decision processes”. arXiv preprint 2017.

---

### Official Review · Reviewer_eVoq · 2026-03-13

**Soundness:** 3
**Presentation:** 2
**Significance:** 3
**Originality:** 3
**Overall Recommendation:** 5
**Confidence:** 4

**Summary:**

The paper studies inverse reinforcement learning (IRL), i.e., the problem of recovering a reward function from expert demonstrations. The authors focus on the maximum causal entropy (MCE) formulation, where the objective is to solve a minimax problem over the reward (r) and the policy ($\pi$) to recover the underlying reward function. The authors propose incorporating a trust-region idea when solving the inner optimization problem ($\max_{\pi}$), by adding a KL constraint that restricts the divergence between the learned policy and the previous policy in order to improve stability.  As a result, the policy update in the inner problem incorporates a Lagrange multiplier associated with the trust-region constraint. Experiments are conducted on several MuJoCo benchmarks and robotics tasks, comparing the proposed method with standard baselines such as GAIL, AIRL, and LSIQ.

**Compliance With Llm Reviewing Policy:**

Affirmed.

**Final Justification:**

The authors’ rebuttal provides substantial additional experiments (e.g., ablation studies on key parameters and comparisons with SOTA methods), which address my concerns to a large extent. My theoretical concerns have also been satisfactorily resolved after carefully reviewing the authors’ responses to both my questions and those of other reviewers.

Overall, it's an excellent rebuttal. I increased my score to **5 (Accept)**.

**Key Questions For Authors:**

1. Can you address my theoretical concerns in Weaknesses 1-3?
2. Can you compare your method with the baselines mentioned in Weakness 2?
3. Since the proposed method appears to require substantial per-task hyperparameter tuning, could the authors clarify whether all baselines were given the same tuning budget?
4. Can you provide ablations for your method parameters in Table 3 as well as the entropy coefficient $\beta$?
5. For reproducibility, could the authors report the full set of tuned hyperparameters for each task?

**Limitations:**

yes

**Strengths And Weaknesses:**

**Strengths:**
- The core idea is generally reasonable. Incorporating trust-region (TR) constraints into policy optimization has been widely used in the reinforcement learning and optimization literature. In this paper, the authors apply a TR constraint to the policy update in the inner optimization of the MCE framework, which could potentially help stabilize the policy learning process.
- The work is built on the maximum causal entropy (MCE) formulation, which was once a well-established and widely studied framework in the imitation learning literature, even though it is now somewhat outdated and known to suffer from stability issues.
- The experimental evaluation appears relatively extensive, including comparisons with several popular baselines and some ablation studies.

**Weaknesses:**
1. The proposed approach is not fully convincing in terms of motivation and execution. For instance, the policy update is derived starting from Eq. (1). However, the objective in Eq. (1) itself appears problematic, since solving it does not necessarily guarantee matching the learned policy with the expert policy. In fact, if the parameter ($\beta$) is small, optimizing (1) may lead the policy to converge to a nearly random policy rather than the expert policy.
2. When presenting the main method, the authors introduce a trust-region constraint defined by the KL divergence between the learned policy and the previous policy in the inner optimization problem. In Lemmas 3.1 and 3.2, the authors derive update forms using Lagrange multipliers associated with the TR constraint. However, applying Lagrangian duality typically requires certain convexity conditions on the optimization problem in Eq. (3). It is not clear whether these conditions hold here, which raises concerns about the correctness and validity of the theoretical results in Section 3.
3. I am also skeptical about the practicality of solving the inner problem in (3) with explicit trust-region constraints. In practice, such constraints are rarely enforced directly. Instead, they are typically handled by introducing a penalty term in the objective function or by applying simpler mechanisms such as clipping or projection to ensure the updated policy does not deviate too far from the previous one. These simpler alternatives should be discussed and compared if the authors want to demonstrate the practical value of their TR-based formulation.
4. The experimental evaluation is missing several recent learning-from-observation baselines, including SMILING [1], SFM [2], and LWAIL [3]. These are strong recent methods and have already shown competitive results on MuJoCo locomotion and manipulation-style tasks, so they would be important comparisons here.
5. I also feel the comparison is not fully fair. According to Table 3 of Appendix C.5, the proposed method has many hyperparameters (roughly six), which implies a substantial tuning effort. Since the method is tuned per task, it is unclear how the baselines are tuned and whether all methods are given the same tuning budget. The paper only mentions tuning learning rates and gradient update counts for each algorithm, but many other algorithm-specific choices can matter significantly for baseline performance. This is particularly relevant for adversarial imitation learning, where prior work [4] has shown that a large number of implementation and tuning choices can strongly affect results.

[1]: Wu, Runzhe, et al. "Diffusing states and matching scores: A new framework for imitation learning." ICLR 2025.

[2]: Jain, Arnav Kumar, et al. "Non-adversarial inverse reinforcement learning via successor feature matching." ICLR 2025.

[3]: Yang, Siqi, et al. "Latent Wasserstein Adversarial Imitation Learning." ICLR  2026.

[4]: Orsini, Manu, et al. "What matters for adversarial imitation learning?." NeurIPS 2021.

---

> ### Author Rebuttal · Authors · 2026-03-31
>
> We thank the reviewer for their thoughtful review and questions. We will start by clarifying our core contribution, which isn’t just a KL-regularized variant of MCE-IRL, but a re-interpretation of how MCE-IRL can be optimized.
>
> MCE-IRL traditionally requires solving a forward RL problem per iteration to obtain the optimal policy for the current reward before performing the dual update. This inner-loop optimization (not instability) is widely recognized as the primary limitation of MCE-IRL and a key obstacle to scaling it to continuous, high-dimensional environments (Ho & Ermon, 2016; Finn et al., 2016; Ren et al., 2024). As a result, many works adopt alternative formulations (e.g., adversarial methods) that avoid this requirement, often at the cost of stability or reward recovery.
>
> We prove that full RL convergence at each iteration is not necessary. Instead, combining (i) a functional dual update, (ii) trust-region (KL-regularized) updates around the previous policy, and (iii) a reward correction step suffices to guarantee monotonic improvement of the dual objective. This yields a more efficient formulation of MCE-IRL without repeated full RL solves.
>
> We therefore respectfully disagree with the low assessment of originality and significance. Our work introduces a new reward correction step and shows that MCE-IRL can be optimized without solving a full RL problem per iteration. This challenges the common assumption that full RL solves are required for MCE-IRL updates. This relaxation enables scaling to challenging tasks like humanoid locomotion, where we outperform established baselines.
>
> We also note that the review assigns a low score for presentation without corresponding comments in the written feedback and would appreciate clarification on any specific concerns.
>
> **New Experiments (anonymous):** https://anonymous.4open.science/r/trirl_rebuttal-304E/
>
> Q1/W1: We consider a standard entropy-regularized distribution matching objective closely related to prior work (e.g. Ziebart et al. 2010; Ho & Ermon, 2016; Ke et al. 2020; Ghasemipou et al. 2020; Kostrikov et al., 2019; Garg et al. 2021). We place the coefficient $\beta$ in front of the divergence term for consistency with prior IRL formulations (Ziebart et al. 2010; Arenz et al. 2016), and hence smaller $\beta$ implies stronger regularization. As in prior work, very small $\beta$ can degrade performance due to over-regularization, reflecting a standard trade-off in entropy-regularized IRL rather than a method-specific issue.
>
> Q1/W2: There is a unique correspondence between policies and occupancy measures (Ho et al., 2016, Prop. 3.1). Rewriting Eq. (3) in occupancy form yields a concave objective: the reward term is linear, the entropy term is concave, and the KL constraint is convex. Hence Slater’s conditions hold and strong duality applies. This implies the trust-region optimal solution is recovered for the optimal multiplier $\eta^\star$ (as used in TRPL). Such KL-constrained policy updates and their duality properties are standard in the trust-region RL literature (Peters et al., 2010; Achiam et al., 2017; Geist et al., 2019 (Proof of Lemma 1)). We will revise the manuscript to clarify this derivation more explicitly.
>
> Q1/W3: Our method does not require strictly enforcing trust-region constraints; the theory also applies when directly optimizing a KL-penalized RL objective. We cannot use PPO-style clipping, since our reward correction step requires access to the Lagrange multipliers. Our implementation already includes both projection (TRPL) and penalty-based variants. We further include GAIL+TRPL as an ablation showing that improvements are not due to KL regularization alone. In addition, we add an ablation of TRIRL without reward correction, which shows that the correction step is crucial in practice. Corresponding results in our anonymous repository.
>
> Q2/W4: We evaluate against established and widely used baselines covering adversarial and non-adversarial approaches as well as reward-learning and occupancy-matching methods (including NEAR, ICLR25). We argue that additional comparisons to IL methods like LWAIL (a concurrent submission) and SMILING (also ICLR25) are nonessential as we already include recent and strong IL baselines despite focusing on IRL. We thank the reviewer for suggesting the IRL method SFM which we still plan to include. Due to time constraints, we prioritized ablations and experiments directly supporting our main claims, particularly on reward learning and scalability.
>
> Q3-Q5: Please refer to our reply to reviewer P7D4 (and anonymous repository).
>
> ---
> Fin et al. "Guided Cost Learning". ICML 2016
>
> Ren, J. et al. "Hybrid Inverse Reinforcement Learning." ICML 2024
>
> Ke L., et al. Imitation learning as f-divergence minimization. WAFR 2020
>
> Peters J, et al. Relative entropy policy search. AAAI 2010
>
> Achiam J, et al. Constrained policy optimization. ICML 2017
>
> Geist M, et al. A theory of regularized markov decision processes. ICML 2019

---

> > ### Author Rebuttal · Reviewer_eVoq · 2026-04-03
> >
> > I thank the authors for their response. However, I find that most of my major concerns remain unaddressed.
> > > *...the review assigns a low score for presentation without corresponding comments in the written feedback...*
> >
> > The presentation and writing are not sufficiently clear. For example, entropy is defined as $H(\pi) = \mathbb{E}_{\rho}[-\log \pi]$, yet in Eq. (3) the term  $\mathbb{E}{\rho}[H]$ appears. This “expectation of an expectation” is confusing and should be clarified.
> >
> > Moreover, the equations use both $\pi$ and $\rho_{\pi}$, which makes the formulation difficult to follow (e.g., when assessing convexity). This complication should be revised.
> > > **Q1/W1: We consider a standard entropy-regularized distribution matching objective closely related to prior work (e.g., Ziebart et al., 2010; ...)**
> >
> > In prior work (e.g., Ziebart et al., 2010 and follow-ups), the entropy regularizer is not combined with a term such as $KL(\rho^{\pi} \| \rho^{E})$. The entropy regularizer is typically introduced to encourage stochastic policies. However, it is effectively equivalent to adding a KL divergence between the learned policy and an uniform (random) policy. As a result, the objective in Eq. (1) can be interpreted as learning a policy that trades off between the expert policy and a random policy, which does not appear to be a well-motivated objective.
> >
> > > **Q1/W2: ...Rewriting Eq. (3) in occupancy form yields a concave objective...**
> >
> > The objective in Eq. (3) does not appear to be concave. The optimization problem involves both $\pi$ and $\rho_{\pi}$, where the policy can be expressed as
> > $$
> > \pi(a \mid s) = \frac{\rho_{\pi}(s,a)}{\sum_{a'} \rho_{\pi}(s,a')}.
> > $$
> > Substituting this into Eq. (3), the resulting objective does not appear to be concave, nor are the constraints clearly convex in either $\pi$ or $\rho$.
> >
> > For example, rewriting the entropy term in terms of $\rho$ yields:
> > $- \rho^{\pi}(s) \cdot \pi(a \mid s) \log \pi(a \mid s)=-\rho^{\pi}(s) \cdot \frac{\rho_{\pi}(s,a)}{\sum_{a'} \rho_{\pi}(s,a')}  \log \frac{\rho_{\pi}(s,a)}{\sum_{a'} \rho_{\pi}(s,a')}.$
> > It is unclear how this expression is concave in $\rho$. Notably, the entropy function is concave in $\pi$, but not necessarily in $\rho$.
> > > **Q2/W4**
> >
> > NEAR, SFM, and SMILING are ICLR 2025 papers; it is therefore unclear why SMILING is considered “non-essential.” I find the authors’ justification here unconvincing. I also do not agree with the statement "the IRL method SFM which we still plan to include. Due to time constraints, we prioritized...". If the central objective of the work is to propose a new IL method and demonstrate superiority over state-of-the-art baselines, omitting such a key comparison weakens the empirical validation.
> >
> > Furthermore, my question regarding baseline tuning stems from concerns about the reported performance of NEAR. As the most recent baseline in the paper, it is unexpected that NEAR underperforms older methods such as GAIL and AIRL across most tasks. This discrepancy raises questions about whether NEAR has been properly tuned or fairly evaluated. The weak performance of NEAR further reinforces the need to include additional recent baselines, as mentioned earlier.
> > >**Q3-5**
> >
> > I read the response to reviewer P7D4 as well as anonymous repository. I have some concerns below:
> >
> > First, conducting an ablation study on the parameters for only a single task is insufficient to properly assess parameter sensitivity.
> >
> > Second, the method appears to require tuning a large number of hyperparameters (approximately 9–11) for each task, making it difficult to identify which parameters are truly critical to performance. If $\epsilon$ and $k$ are the key parameters, it is unclear why they are not tuned per task while keeping the remaining parameters fixed.
> >
> > Additionally, you mention that you use wandb to tune all methods. To my knowledge, W&B Sweeps support analyzing parameter importance. Can you use this feature to provide a more detailed parameter analysis?
> >
> > **I also have two follow-up questions.**
> > - According to Figure 4, different tasks use different training budgets ranging from 3e7 to about 4e8 gradient steps. This is uncommon in IL experimental setups, where methods often use a fixed training budget across tasks. A common choice is 1e6 gradient steps, which is used in many recent IL works such as LWAIL, SFM, and SMILING. Could you explain the motivation behind your training budget setting?
> >
> > - The convergence speed of your method appears to be relatively slow. For example, it takes 2e7–4e7 gradient steps to converge on standard MuJoCo locomotion tasks. However, results in LWAIL show that both LWAIL and its baselines can converge within 1e6 steps on the same tasks. Could you explain the large gap in convergence speed between your method and LWAIL?
> >
> > **Overall, the critical concerns remain unresolved, and I therefore maintain my original evaluation.**

---

> > > ### Author Response · Authors · 2026-04-04
> > >
> > > ## Presentation (will be fixed in a revision)
> > > - Thank you for this feedback. Our original submission was indeed slightly inconsistent by defining $H(\pi)$ in Sec. 2 as the causal entropy, but in some places using an unnecessary expectation $E_{\rho\_\pi(s)} [H(\pi)]$. Generally, this expectation over an entropy is common in RL/IRL (e.g., Fu et al., 2018, Section 3;  Haarnoja et al. 2018, Eq. 1) and is useful for being explicit about the state distribution when using $H(\pi)$ for the conditional entropy. Though our redundancy can be confusing, we emphasize that the notation is mathematically correct and disambiguous. Therefore, we argue that this is a minor issue.
> > > - Using $\pi(a|s)$ for the policy, and $\rho\_{\pi}(s)$ / $\rho\_{\pi}(s,a)$ for its occupancy/their product is standard notation in the field. Notable examples include Ho et al. (2016) and Garg et al. (2021), who also used this notation for their entropy regularized divergence-minimization objective $\min_{\pi} d_{\psi}(\rho_{\pi}, \rho_{E}) - H(\pi)$.
> > >
> > > ## Q1/W1 and Q1/W2
> > > Please refer to our new reply to reviewer CjCM. We emphasize that both Q1/Q2 can be addressed by considering several prominent and well-cited prior works in IL and entropy regularized RL. **We will revise our submission to better discuss the theoretical foundations of our work, but kindly request the reviewer to reevaluate their concerns after considering the mentioned prior work**.
> > >
> > > (i) Causal entropy regularizes toward simpler behavior, imposes few assumptions on the policy, and has well-known practical advantages such as increased robustness (Eysenbach et al., 2021). That strong regularization could result in random policies is an inherent effect of regularization and **not a specific limitation of our method.** Entropy regularized divergence minimization (exactly like our objective) was used by GAIL (Eq 15: minimizing entropy-regularized JS divergence), IQ-Learn (several entropy-regularized divergences), AIRL, and the work that underlies our submission (Arenz et al. 2016, Eq. 6).
> > >
> > > (ii) The strong duality of the entropy regularized TR RL problem has been widely studied. Expected entropy and expected KL under $\rho(s)$ are commonly used in RL and IL/IRL, and their concavity/convexity is well-established.
> > > - **(Ho et al. 2016 Lemma 3.1) and (Neu et al. 2017, Appendix A1) clearly dispel the reviewer’s concerns that “rewriting the entropy in terms of $\rho$ yields …”.**
> > > - In our new response to reviewer CjCM, we show that the objective’s terms are concave (entropy) and linear (reward), with convex (KL) and linear (flow) constraints. (Ho et al. 2016) also make it clear that “Prop/Lemma 3.1 allow us to freely switch between policies and occupancy measures”.
> > >
> > > ## Q3 - Q5
> > > ***UPDATE: The anonymous repo has been updated with ablations (new tasks), w&b sweep stats, and the new baseline SFM***. Please also see the data in our new response to reviewer P7D4.
> > >
> > > Hyperparameters: Different environments naturally have different observation/action spaces and different exploration requirements. Keeping learning rates, gradient steps etc. fixed for all environments is typically non-standard, and secondary parameters are also tuned (e.g, IQ-Learn Appendix D.1.2 mentions different learning rates, ent coeffs, etc., for different tasks even though these are secondary hyperparams). We follow the same standard methodology.
> > >
> > > Baselines: Our main aim is to propose a new IRL method (not IL). **NEAR (ICLR25) and SMILING (ICLR25) are both very similar methods that use the same score-based reward-learning scheme**. We chose NEAR because it claims to learn global rewards. LWAIL (ICLR26) was accepted after our submission and considers a different setting of IL from observation. SFM is an IRL method, and is now included in our rebuttal (see anonymous repository).
> > >
> > > NEAR Performance: This is not a tuning issue. We put significant effort into tuning NEAR, using sweeps and manual tuning afterwards. NEAR probably performs poorly because we use fewer expert demos than were used in the original work (**learning score-based rewards needs lots of data, Appendix C.1**). Further, we remind the reviewer that **NEAR does not always outperform baselines even in the original work (typically being on par with AMP)**.
> > >
> > > **Follow-Up Q1/Q2:**
> > > - Using the same number of transitions for both easy/hard tasks would be wasteful. It is sensible (and common practice) to select the number of steps depending on the environment complexity. **This is consistent with the experiments in NEAR (Table 5)**.
> > > - The large training steps are a result of using massively parallel RL environments (Appendix C.2). In doing so, we obtain several thousand steps/second, which leads to much faster wall-clock times. However, doing so trades off sample efficiency (as can also be seen in NEAR). In general, **this is not a reliable indicator of our method’s convergence speed**.
> > >
> > > ---
> > >
> > > Neu et al. 2017. A unified view of entropy-regularized MDPs.  2017.

---

### Decision · Program_Chairs · 2026-04-30

**Decision:**

Accept (regular)

**Comment:**

The submission initially received highly divergent reviews, primarily centered on the theoretical validity of the objective and the practical necessity of the reward correction step. Through an extensive rebuttal process, the authors successfully addressed these concerns. All the reviewers give clear accept scores (5 Accept). AC has checked the submission, the reviews, the rebuttal, and the discussion. The paper provides a solid bridge between local policy optimization and global reward optimality. Given the resolution of the initial reviewer disagreements and the clear strength of the empirical results on challenging robotics benchmarks, the AC recommends acceptance.